# Lipid order and charge protect killer T cells from accidental death

Jesse A. Rudd-Schmidt[1,2,9], Adrian W. Hodel[3,4,9], Tahereh Noori[1], Jamie A. Lopez[1,8], Hyun-Jung Cho[5], Sandra Verschoor[6], Annette Ciccone[6], Joseph A. Trapani[2,6], Bart W. Hoogenboom ●[3,4,7,10]* & Ilia Voskoboinik[1,2,10]*

Killer T cells (cytotoxic T lymphocytes, CTLs) maintain immune homoeostasis by eliminating virus-infected and cancerous cells. CTLs achieve this by forming an immunological synapse with their targets and secreting a pore-forming protein (perforin) and pro-apoptotic serine proteases (granzymes) into the synaptic cleft. Although the CTL and the target cell are both exposed to perforin within the synapse, only the target cell membrane is disrupted, while the CTL is invariably spared. How CTLs escape unscathed remains a mystery. Here, we report that CTLs achieve this via two protective properties of their plasma membrane within the synapse: high lipid order repels perforin and, in addition, exposed phosphatidylserine sequesters and inactivates perforin. The resulting resistance of CTLs to perforin explains their ability to kill target cells in rapid succession and to survive these encounters. Furthermore, these mechanisms imply an unsuspected role for plasma membrane organization in protecting cells from immune attack.

[1] Killer Cell Biology Laboratory, Peter MacCallum Cancer Centre, 305 Grattan Street, Melbourne, VIC 3000, Australia. [2] Sir Peter MacCallum Department of Oncology, University of Melbourne, Melbourne, VIC 3000, Australia. [3] London Centre for Nanotechnology, University College London, 19 Gordon Street, London WC1H 0AH, UK. [4] Institute of Structural and Molecular Biology, University College London, Gower Street, London WC1E 6BT, UK. [5] Biological Optical Microscopy Platform, The University of Melbourne, Parkville, VIC 3010, Australia. [6] Cancer Cell Death Laboratory, Peter MacCallum Cancer Centre, 305 Grattan Street, Melbourne, VIC 3000, Australia. [7] Department of Physics and Astronomy, University College London, Gower Street, London WC1E 6BT, UK. [8] Present address: Bristol-Myers Squibb, 4 Nexus Ct, Mulgrave, VIC 3170, Australia. [9] These authors contributed equally: Jesse A. Rudd-Schmidt, Adrian W. Hodel. [10] These authors jointly supervised the work: Bart W. Hoogenboom, Ilia Voskoboinik. *email: b.hoogenboom@ucl.ac.uk; ilia.voskoboinik@petermac.org

The immune system relies heavily on killer T cells (cytotoxic T lymphocytes, CTLs) to eliminate virus-infected or cancerous cells. A CTL binds to, and forms an immunological synapse with its target, then secretes a $Ca^{2+}$-dependent pore-forming protein (perforin) and a cocktail of pro-apoptotic serine proteases (granzymes) into the synaptic cleft[1,2]. Perforin-mediated membrane disruption is indispensable for allowing granzymes access to key substrates in the target cell cytosol, enabling the initiation of apoptosis of dangerous cells and to maintain immune homoeostasis. Consequently, failure to release functional perforin results in fatal immune dysregulation or increased susceptibility to viruses and to haematological cancers[1,3,4].

In a first and essential step, secreted perforin interacts with a target cell via $Ca^{2+}$-dependent membrane binding, mediated through its C2 domain. This is followed by oligomerization into short and non-inserted prepore assemblies and, ultimately, by insertion and further assembly of an entire mature pore across the plasma membrane[5–7]. Although the CTL and the target cell plasma membranes are both exposed to perforin within the synapse, paradoxically, only the target cell membrane is disrupted[8]. As a consequence, granzymes will penetrate and kill the target cell, whereas the CTL is almost invariably spared.

A number of mechanisms have been advanced to account for this apparently unidirectional action of perforin. Early reports indicated that perforin binds to specific lipid headgroups[9], but subsequent work suggested tight lipid spacing to be a more relevant factor, making lymphocytes inherently resistant to perforin[10]. Yet CTLs can be killed by fratricide, provided that they can present cognate peptide on MHCI[8]. Other explanations advanced to date include cleavage and inactivation of secreted perforin by the constitutive lysosomal cysteine protease cathepsin B (CatB)[11], but this hypothesis was also refuted as CTLs from CatB-null mice survive multiple successive interactions with target cells with the resilience of CatB-competent counterparts[12]. In addition, some lymphocyte protection may potentially be acquired through externalization of an abundant cytotoxic

granule transmembrane protein, LAMP-1[13]. These various explanations should be considered in the context of recent single-cell analyses, which revealed that more than 95% of killer lymphocytes survive their encounter with a single target, ensuring their ability to disengage, then engage again to successively kill other dangerous cells[8,14]. This result highlights the extraordinary effectiveness of the mechanism(s) underpinning the target-cell specificity of, and/or lymphocyte protection against perforin in the immune synapse.

In this study, we use a variety of biochemical, biophysical and cell biological approaches to resolve the critical and long-standing question of the unidirectional action of perforin in the immunological synapse. We demonstrate that the CTL protects itself from perforin by dynamic control of its membrane lipid composition. Specifically, we identify two protective mechanisms: the CTL membrane repels perforin by arranging its lipids in a higher-ordered state (lipid rafts) at the immunological synapse, while also exposing phosphatidylserine (PS) within the synapse, thus creating a negatively charged sink that sequesters and inactivates any residual perforin. The resistance of CTLs to perforin underpins their capacity to kill multiple targets, and enables them to maintain immune homoeostasis.

## Results

**CTLs are resistant to perforin binding and lysis.** We initially explored whether killer lymphocytes are intrinsically more resistant to perforin than their targets, and found that both primary activated CTLs (CD8+ T cells from BL/6 OTI transgenic mice) and natural killer cells (isolated from BL/6 mice) required 10s- to 100s-fold more recombinant perforin to achieve the same level of lysis as common target cell lines (Fig. 1a). This suggests that perforin binds less efficiently to killer lymphocyte membranes and/or that its pore-forming function is impaired on these membranes.

To assess perforin binding levels without the confounding effect of cell lysis, we used a non-lytic mutant of perforin (TMH1-PRF) that binds as efficiently to target membranes as wild-type

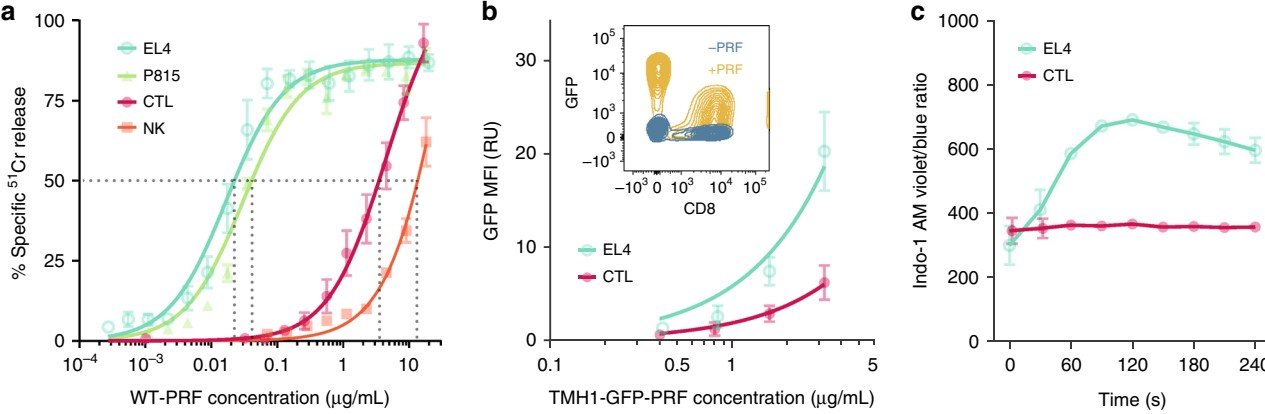

**Fig. 1** CTLs bind less perforin and resist pore formation. **a** $^{51}$Cr release cytotoxicity assay on CTLs, natural killer (NK) cells, and EL4 and P815 target cells upon exposure to recombinant WT-PRF. CTLs are 2–3 orders of magnitude more resistant than target cells, as assessed at 50% $^{51}$Cr release (dotted lines). Curves represent Michaelis–Menten fits to the data. **b** Binding of TMH1-GFP-PRF to a 1:1 mixture of CTL and EL4 cells, as assessed by flow cytometry, with GFP geometric mean fluorescence intensity (MFI) measured in relative units (RU). The cells were stained with anti-CD8-allophycocyanin (APC) prior to flow cytometry to differentiate the two cell types. The comparison was made for CTL and EL4 cells of the same size, as derived from forward and side scatter. Curves represent Michaelis–Menten fits to the data. Inset: example of flow cytometry data with TMH-GFP-PRF staining of a 1:1 mixture of CD8+ (CTL) and CD8− (EL4) cells. **c** $Ca^{2+}$ influx into Indo-1 AM labelled CTL and EL4 cells upon exposure to sub-lytic amounts of WT-GFP-PRF. Membrane perforation by perforin was measured via the $Ca^{2+}$ influx over time, detected as an increase in the ratio of violet (400 nm) to blue (475 nm) fluorescence emission; WT-GFP-PRF was added at $t = 0$. See Supplementary Fig. 1b, c for control experiments that demonstrate similar levels of perforin binding to both cell types, as well as the specificity of the observed difference to perforin. Curves provide a visual guide connecting data points. Throughout the panels, each data point represents a mean ± standard error of mean (s.e.m.) of three independent experiments. Source data for all panels are provided as a Source Data file.

perforin (WT-PRF), but incorporates an engineered disulphide bond that tethers the transmembrane domain and prevents its insertion into the target cell membrane[6]. Upon binding to the membrane, TMH1-PRF instead forms short prepore oligomers on the membrane, which represent a transient, intermediate state of WT-PRF pore assembly[6]. GFP-tagged variants (WT-GFP-PRF[15] and TMH1-GFP-PRF) enabled us to measure perforin binding to cells directly by fluorescence. As previously shown for TMH1-PRF[6], the pore forming and lytic functions of TMH1-GFP-PRF were restored by reducing the engineered disulphide bond with dithiothreitol (DTT, Supplementary Fig. 1a). Using TMH1-GFP-PRF, we found that CTLs (CD8[+]) indeed bound less perforin than EL4 (CD8[−]) target cells of the same size (Fig. 1b, Supplementary Fig. 2a).

Although lower than the observed binding levels for target cells, perforin binding to CTLs is far from negligible (Fig. 1b), yet their plasma membranes almost invariably withstand disruption, as shown here and elsewhere[8]. Moreover, when the concentration of WT-GFP-PRF was adjusted so that CTLs and target cells bound similar amounts of perforin to their membranes, the CTL membranes still remained intact, while the target cell membranes were perforated, as indicated by $Ca^{2+}$ flux (Fig. 1c, Supplementary Figs. 1b, c, 2b). These results were confirmed further by demonstrating the survival of CTLs despite their binding significant amounts of WT-GFP-PRF (Fig. 2a, −EGTA, Supplementary Fig. 2a; note the significant GFP[+] population for CD8[+] cells at 37 °C).

We next (Fig. 2a, +EGTA) compared this CTL-bound perforin with the two known assembly states of perforin. In the early stages of pore formation, non-porating, prepore assembly intermediates are formed by WT-PRF at lower temperature and by TMH1-PRF in the absence of DTT[6]. These prepores can be removed from the membrane by depleting free $Ca^{2+}$ by the addition of EGTA. After binding, WT-PRF can form functional, transmembrane pore assemblies on target cells and model membranes at 37 °C. These perforin pores are EGTA resistant[16]. Intriguingly, although not lytic, the CTL-bound WT-GFP-PRF resembled functional, membrane-inserted perforin by being resistant to EGTA (Fig. 2a, +EGTA, Supplementary Fig. 2a). This is in contrast to the EGTA sensitivity shown by the non-lytic TMH-GFP-PRF (Fig. 2b; for quantification of all conditions, see Fig. 2c). Taken together, these observations indicate that CTLs have an additional layer of protection, where membrane-bound perforin remains non-functional (i.e., non-porating) despite being apparently inserted into the membrane.

To explain both the reduced membrane binding and the reduced functionality of membrane-bound perforin on CTLs, we hypothesized that CTL plasma membranes may be more resistant to perforin binding and pore formation due to their lipid composition[10]. This hypothesis was also motivated by the observations that perforin does not require partner proteins for its membrane binding and pore formation; that it has no known inhibitors in the CTL membrane[12]; and that our experiments (Supplementary Figs. 2c, 3) did not confirm a previously postulated protective role of cytotoxic granule membrane protein CD107a (LAMP-1)[13].

**High lipid order protects membranes from perforin binding**. To assess perforin binding and pore formation on membranes of defined lipid composition, we used atomic force microscopy (AFM) on supported lipid bilayers[6,17–20]. Ternary mixtures of cholesterol, sphingomyelin (SM) and (liquid-phase) phosphatidylcholine (PC) provided general biophysical model systems for the plasma membrane in which to assess the effect of lipid order[21–23]. By systematically varying membrane order across

single-phase and phase-separated regimes of this model membrane system (Supplementary Fig. 4), we found that perforin (specifically: WT-PRF) readily forms pores on PC-rich, liquid-disordered lipid phases, but avoids SM/cholesterol-rich liquid-ordered domains (Fig. 3a, b). Overall, higher SM and cholesterol content caused the membranes to be increasingly refractory to perforin pore formation (see dark areas in Fig. 3a), coinciding with regimes of enhanced lipid order[21–23]. Membrane binding of the non-porating TMH1-PRF mutant followed the same pattern (Supplementary Fig. 5), demonstrating that the observed reduction in WT-PRF pore formation on SM/cholesterol-rich membranes was due to a reduction in perforin binding. When more disorder-prone variants of SM (18:1 SM[24]) and of cholesterol (7-ketocholesterol, 7KC[25]) were used, perforin binding and pore formation were restored (Fig. 3c, d, Supplementary Fig. 6). Hence increased lipid order causes membranes to be refractory to perforin binding (and thus also to pore formation).

To test the effect of membrane order in live cells[26], we reduced membrane order in CTLs by loading them with increasing quantities of 7KC[25]—thus substituting cholesterol in the membrane—prior to perforin exposure (Fig. 4a). Of note, a non-toxic concentration of 7KC (as used here) did not sensitize the cells to osmotic stress (Fig. 4b, c, Supplementary Fig. 2d). As predicted by our AFM findings, increased disorder in the CTL plasma membrane led to an increase of TMH1-GFP-PRF binding (Fig. 4d, Supplementary Fig. 2e) and accordingly to an increase in lysis by WT-PRF (Fig. 4e, Supplementary Fig. 2d), compared with cholesterol-treated cells. Remarkably, thus treated CTLs had their perforin sensitivity increased by up to tenfold, to levels comparable with untreated target cells (Fig. 4e). This increased sensitization to perforin was also observed for primary natural killer cells (Fig. 4f), but less so for target cells (Fig. 4e). We conclude that enhanced lipid order in the CTL plasma membrane reduces the efficiency of perforin binding. Consistent with this conclusion and the differences in perforin sensitivity between CTLs and their targets (Figs. 1, 2), we also note that CD8[+] T cells show over 30-fold higher staining for lipid raft marker GM1 than target cells (Supplementary Fig. 7).

**PS inactivates membrane-bound perforin**. Having determined a lipid-based mechanism that protects CTLs from perforin binding, we next set out to further explore our surprising observation of non-functional, yet membrane-bound and possibly inserted perforin on the CTLs (Figs. 1b, c, 2). To this end, we expanded the AFM analysis to phospholipids that are typically located in the inner leaflet of the plasma membrane[27]. Surprisingly, we found that perforin efficiently bound to membranes composed of PS, but did not form the well-defined, static arc- or ring-shaped assemblies that are characteristic[5,6,19] of transmembrane pores (Fig. 5a). Instead, WT-PRF and DTT-unlocked TMH1-PRF formed static protein clusters on PS, which sat up to ~5 nm higher above the membrane surface than locked (−DTT) TMH1-PRF prepores on both PC and PS. These static clusters were also higher (by the same amount) than functional perforin (WT-PRF and TMH1-PRF +DTT) on PC membranes. Fully consistent with our observations on CTL-bound perforin (Fig. 2a), the perforin clusters on PS membranes were resistant to $Ca^{2+}$ depletion by EGTA (Fig. 5b). In binary PC/PS mixtures, cluster formation was increased and functional pore formation reduced upon increasing PS content (Fig. 5c). We explored these unexpected observations further by assessing the behaviour of perforin on model membranes doped with other negatively charged lipids, specifically: DOPG (1,2-dioleoyl-sn-glycero-3-phospho-(1′-rac-glycerol)) or CS (cholesterol sulphate). Similar to PS, perforin pore assembly was impaired on both membranes (Supplementary Fig. 8). These

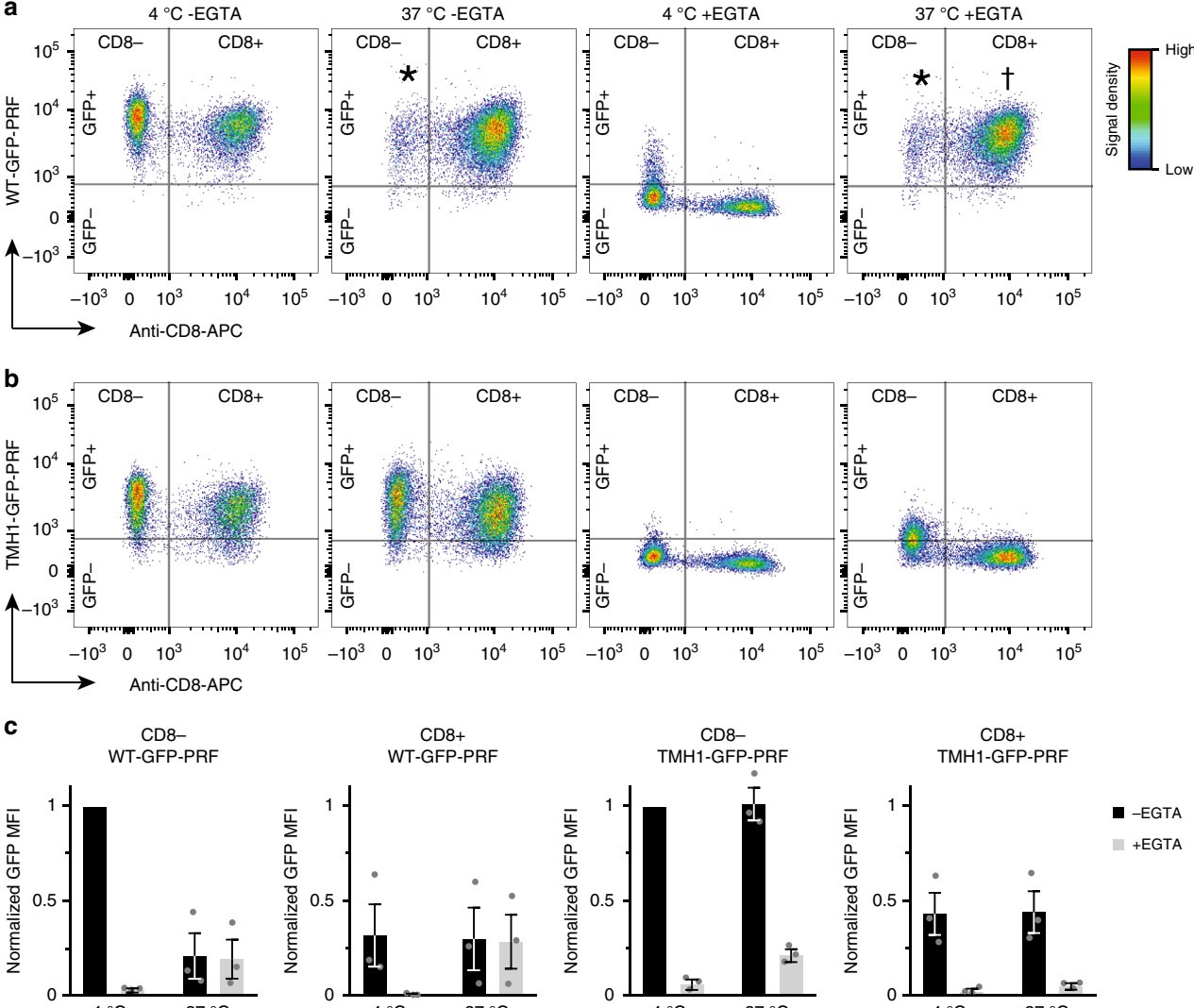

**Fig. 2** Residual perforin bound to CTLs is deactivated. **a** Binding of WT-GFP-PRF to EL4 (CD8−) and CTL (CD8+) cells, at concentrations that are sub-lytic to CTLs, as assessed by flow cytometry. At 4 °C and in the presence of $Ca^{2+}$ (−EGTA), both cell types bind perforin (GFP+), but the CTLs at a lower level than EL4 cells. At 37 °C and in the presence of $Ca^{2+}$ (−EGTA), WT-GFP-PRF is lytic to the EL4 cells, so no significant CD8− population remains under those conditions (asterisks). Upon subsequent $Ca^{2+}$ chelation by 2 mM EGTA, non-porating perforin assemblies (WT-GFP-PRF at 4 °C) are removed from the cell membranes (GFP−). However, $Ca^{2+}$ chelation does not affect WT-GFP-PRF bound to CTLs at 37 °C (GFP+, dagger), despite WT-GFP-PRF being apparently non-porating (see Fig. 1c). **b** Binding of non-lytic TMH1-GFP-PRF to EL4 (CD8−) and CTL (CD8+) cells, as assessed by flow cytometry. In the presence of $Ca^{2+}$ (−EGTA), TMH1-GFP-PRF binds to both cell types, but less to CTLs, as observed for WT-GFP-PRF. Upon subsequent $Ca^{2+}$ chelation by 2 mM EGTA (+EGTA), TMH1-GFP-PRF is invariably removed from the cell membranes, leaving only GFP− populations (both at 4 °C and 37 °C). **c** Quantification of flow cytometry results, with MFI of WT-GFP-PRF and TMH-GFP-PRF signals, normalized to the respective MFIs for the CD8− population at 4 °C. EGTA exposure drastically reduces the GFP MFI for all cases at 4 °C and for (non-lytic) TMH1-GFP-PRF at 37 °C. By contrast, WT-GFP-PRF is EGTA resistant once bound to CD8+ cells at 37 °C. Note that the low GFP MPI at 37 °C in the first panel (CD8−, WT-GFP-PRF) is due to the majority of the CD8− cells being killed and thus not yielding a GFP signal in the flow cytometer anymore. The TMH1-GFP-PRF data (−EGTA) allow for a quantitative comparison between binding to CD8+ and CD8− cells, showing that the CTLs (CD8+) bind less than half as much perforin as the target cells (CD8−). Each column represents a mean (±s.e.m.) of three independent flow cytometry experiments. Source data for **c** are provided as a Source Data file.

results show that the abnormal behaviour of perforin on PS membranes, i.e., the formation of non-porating assemblies, is due to the negative charge of the membrane surface.

On closer inspection by negative-stain electron microscopy on PS monolayers (Fig. 6a), the perforin clusters were found to consist of intermediates that appear similar to prepore (TMH1-PRF) assemblies in their size and subunit spacing (for quantification, see Fig. 6b, c), albeit more aggregated. Specifically, the subunit spacing of the clustered assemblies is $3.8 \pm 0.8$ nm, compared with $3.7 \pm 0.7$ nm for the locked TMH1-PRF as measured on PS here, and with $3.86 \pm 0.13$ nm for locked

TMH1-PRF and with $2.55 \pm 0.09$ nm for functional perforin pores (WT-PRF and TMH1-PRF +DTT) as previously determined on PC-rich membranes[6]. Taken together, these observations on PS are consistent with perforin assembly that is trapped in a dysfunctional (dead-end) state that prevents it from completing the prepore-to-pore transition.

If our hypotheses are correct, we would expect perforin to colocalize with exposed PS[28] on the CTL plasma membrane. To test this, we performed confocal microscopy on activated CTLs that were pre-treated with annexin V-Alexa Fluor 568, as a marker for PS exposed on the cell surface[29]. Importantly, this pre-

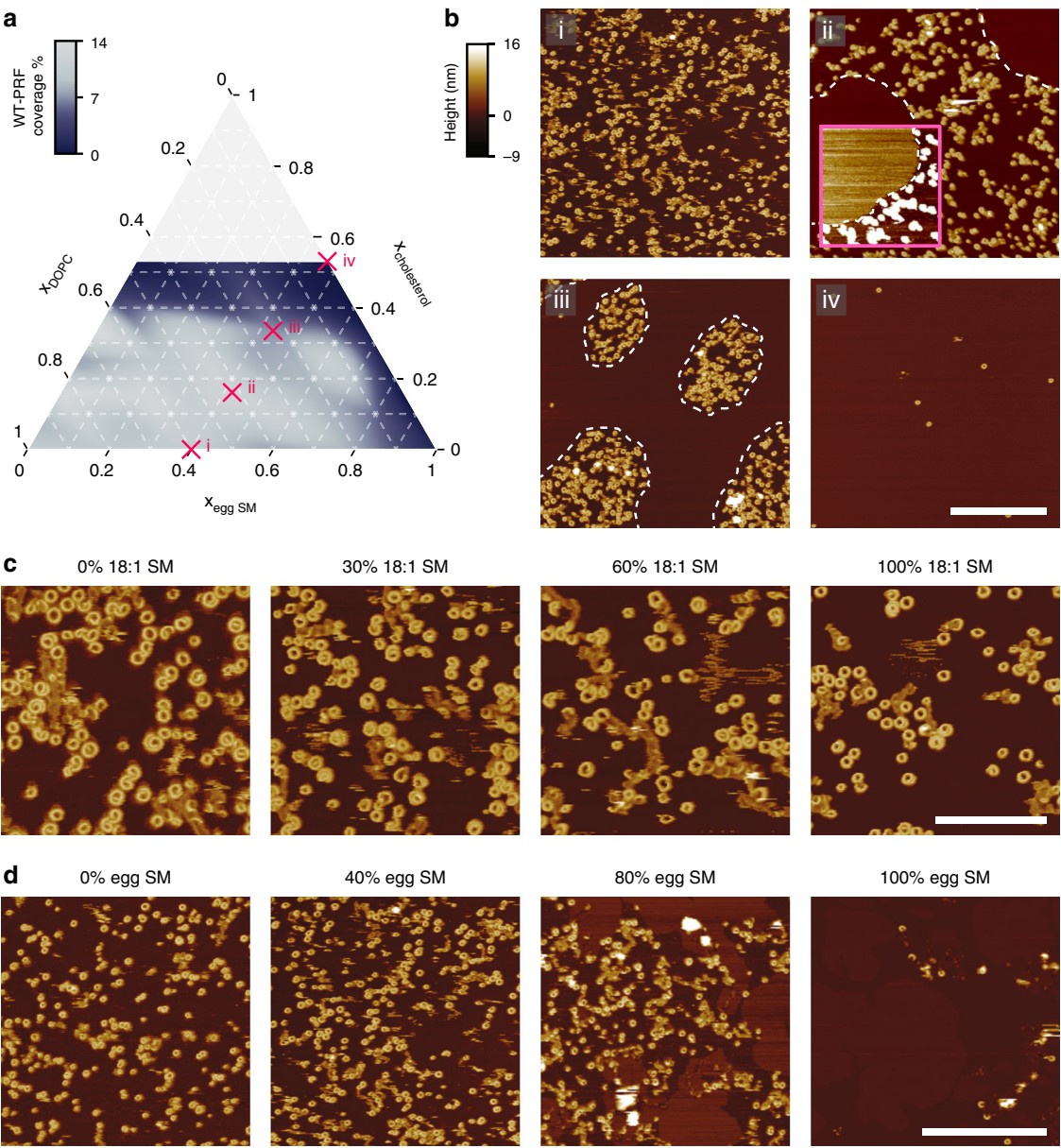

**Fig. 3** Lipid order protects membranes from perforin pore formation. **a** Perforin coverage (as % of membrane surface, as assessed by AFM imaging) after incubation of supported DOPC/egg SM/cholesterol bilayers with 150 nM WT-PRF, as a function of the molar fractions ($x$) for the constituents of the bilayer (total = 1). See Supplementary Fig. 4 for explanation of the lipid phases observed in these membranes. Source data are provided as a Source Data file. **b** AFM images of perforin pores on supported lipid bilayers of the compositions i–iv labelled in **a**. AFM samples were incubated and imaged at 37 °C. For the area marked by a pink square in the top right image, the colour scale has been saturated (4 nm instead of 25 nm full range) to more clearly identify the enhanced thickness (height) of a liquid-ordered domain and the absence of perforin pores (white at this scale) on such domains. Throughout the images, lipid-phase boundaries are marked by dashed lines. Scale bar, 500 nm. **c** Representative AFM scans of supported DOPC bilayers, doped with different relative amounts of 18:1 SM (0–100%), and incubated with 150 nM WT-PRF. The doping with this disorder-prone SM variant does not appear to affect perforin pore formation, showing a similar coverage across all samples. This demonstrates that the observed perforin inhibition on egg SM membranes in **a** and **b** is not due to the SM headgroup (identical for egg SM and 18:1 SM), but due to the SM-induced changes in membrane order; unlike egg SM, pure 18:1 SM is in a liquid disordered state at 37 °C, just like DOPC[24]. Colour scale as in **b**. Scale bar, 200 nm. **d** As **c**, for comparison, except that the DOPC bilayer was doped with egg SM (0–100%). Phase separation is visible at 80%, and perforin coverage is notably diminished at 100% egg SM. These images were extracted from the dataset used to compose the plot shown in **a**. Note the different length scale. Colour scale as in **b**. Scale bar, 500 nm. All samples were incubated and imaged at 37 °C.

labelling with annexin V ensured detection only of PS already exposed on the CTL membrane before perforin addition, thus avoiding any contribution of PS flip/flop upon perforin association with the plasma membrane[30]. WT-GFP-PRF colocalized with punctate regions corresponding to (non-apoptotic) externalized PS (Fig. 7a, b). These punctate regions were also enriched for

the lipid-raft marker cholera toxin B (labelled GM1 in Fig. 7a, b), as was previously reported for both CTLs[31] and B cells[32]. To directly investigate the efficacy of this protective mechanism, WT-GFP-PRF was added to a 1:1 mixture of EL4 cells and CTLs in the presence of annexin V-Alexa Fluor 647 to identify exposed PS (Fig. 7c). EL4 cells bound perforin uniformly around their

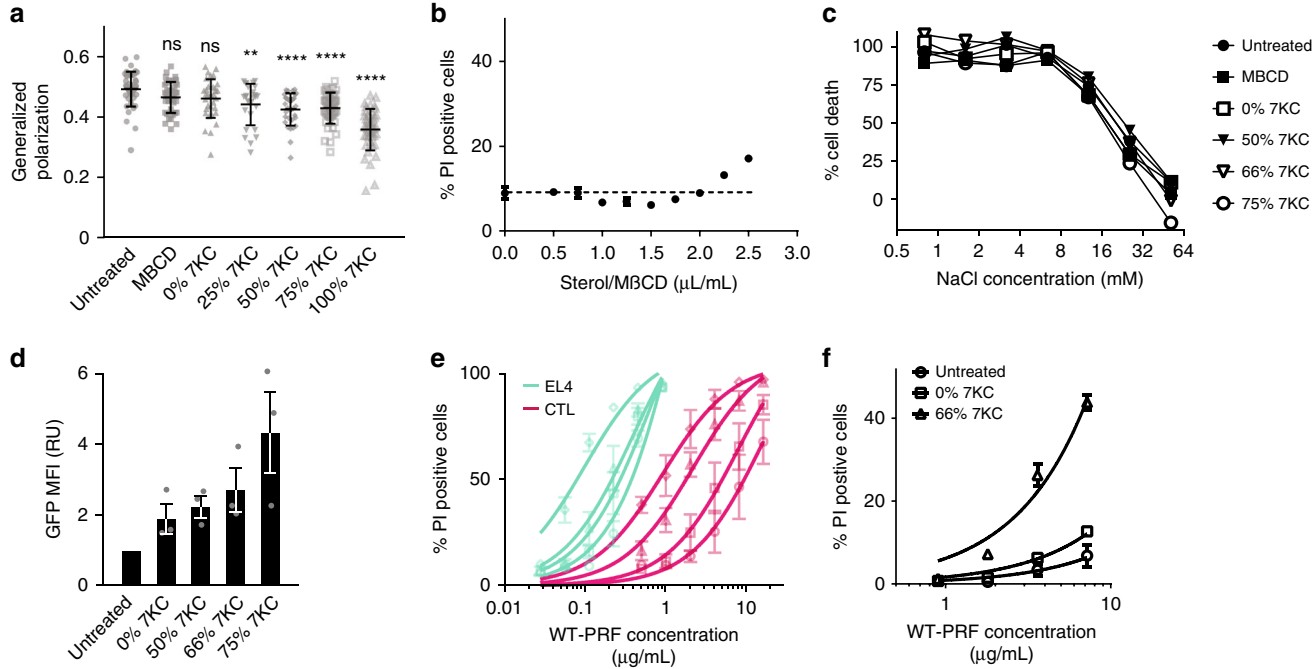

**Fig. 4** Reduced membrane order sensitizes cells to perforin lysis. **a** Lipid order in the CTL plasma membrane for cells treated with different ratios of 7KC/cholesterol, as measured by GP values[26] (see Methods). Data points represent individual cells, measured in three independent experiments and are overlaid with mean ± s.e.m.; statistical significance was assessed using ANOVA with Dunnett's post-hoc analysis, where 'ns'–not significant, **$p < 0.01$, ****$p < 0.001$. **b** Determination of a non-cytotoxic concentration of 7KC. A total of $5 \times 10^5$ CTLs were treated with increasing amounts of a 75% 7KC/25% cholesterol mix in MßCD, and cell death assessed by propidium iodide (PI) staining. A total of 2.25 µL/mL was chosen as the optimal concentration for following cholesterol loading experiments. Data points show mean ± s.e.m. of three independent experiments with the dotted line marking background cell death (no sterol/MßCD added). **c** 7KC treatment of CTLs does not sensitize cells to osmotic stress. $^{51}$Cr-loaded CTLs were treated with varying ratios of 7KC/cholesterol, incubated at different concentrations of NaCl, and cell death assessed by $^{51}$Cr release. Data points show mean of two independent experiments, with three technical replicates each. **d** Perforin binding to CTLs pre-loaded with different ratios of 7KC/cholesterol. The GFP MFI of cells that had been exposed to TMH1-GFP-PRF at 37 °C was measured by flow cytometry and normalized to the MFI of untreated cells. The data represent mean ± s.e.m. of three independent experiments. **e** The reduced membrane order of 7KC-treated CTLs (red) and EL4s (blue) sensitizes the cells to WT-PRF lysis, as determined by PI staining of untreated cells (circles) and cells loaded with 50% (squares), 66% (triangles) and 75% (diamonds) 7KC. Data points show mean ± s.e.m. of three independent experiments. Curves represent Michaelis–Menten fits. **f** Reduced membrane order of 7KC-treated murine NK cells sensitizes the cells to WT-PRF lysis. Data show mean PI staining of untreated cells (circles) and cells loaded with 0% (squares) and 66% (triangles) 7KC. Error bars represent s.e.m. ($n = 3$ for all points except highest perforin concentration where $n = 2$). Source data for all panels are provided as a Source Data file.

periphery and were lysed within minutes, whereas on CTLs, perforin signal was strongly associated with distinct regions where PS was exposed (Fig. 7b, c; Supplementary Video 1). Of note, nucleated cells employ an exocytic membrane repair response that is capable of protecting cells from an acute perforin lysis[15], meaning EL4 cells do not immediately become annexin V positive, at least not for the perforin concentrations used here. However, over time, more EL4 cells die from perforin-induced damage, as highlighted in Supplementary Video 2.

Overall, inefficient perforin binding to CTLs (Fig. 1b), coupled with its inactivation by the plasma membrane PS, provides an explanation for their resistance to perforin lysis (Supplementary Video 2, Supplementary Fig. 9). Linking the results of AFM and cell-based experiments, it is important to note that the effective exposure of PS in the outer leaflet of our homogeneous supported bilayer can be lower than the overall PS concentration by more than a factor of 2, due to interaction of the charged PS headgroups (in the inner leaflet) with the mica substrate[33]. While this does not affect the overall trend observed for increased PS contents, it implies that the externalized PS in the cell membrane can be at levels well below 60% (observed in Fig. 5c) to achieve the same inhibitory effect on perforin pore assembly.

Taken together, we conclude that killer lymphocytes are protected in at least two ways against the perforin they secrete:

the lipid ordering in their membranes acts as a deflective shield against perforin binding and, in addition, perforin is scavenged and neutralized via the formation of non-lytic perforin clusters due to (negatively charged) PS that is externalized on the lymphocyte pre-synaptic membrane.

To consider this in the context of the immune synapse, we note that SM- and cholesterol-rich lipid domains (rafts) dynamically re-arrange and merge at the CTL membrane during immune synapse formation, as extensively demonstrated by others[34–36]. Hence, the presynaptic membrane will have a further increased lipid order compared with the overall CTL membranes tested here, thus enhancing its protection against perforin binding. In addition, PS was found to be exposed on T-cell membranes and directed to the presynaptic membrane[31,37,38] (Fig. 7d; Supplementary Video 3), allowing PS to act as a sink that binds and inactivates any perforin reaching the CTL membrane in the immune synapse (Fig. 8).

## Discussion

Unregulated binding and pore formation by perforin on the plasma membrane of CTLs and other cytotoxic lymphocytes would make immune killer cells as vulnerable to their own secreted potent cytotoxins as a target cell, and greatly reduce the efficiency of a cytotoxic response to dangerous pathogens. For

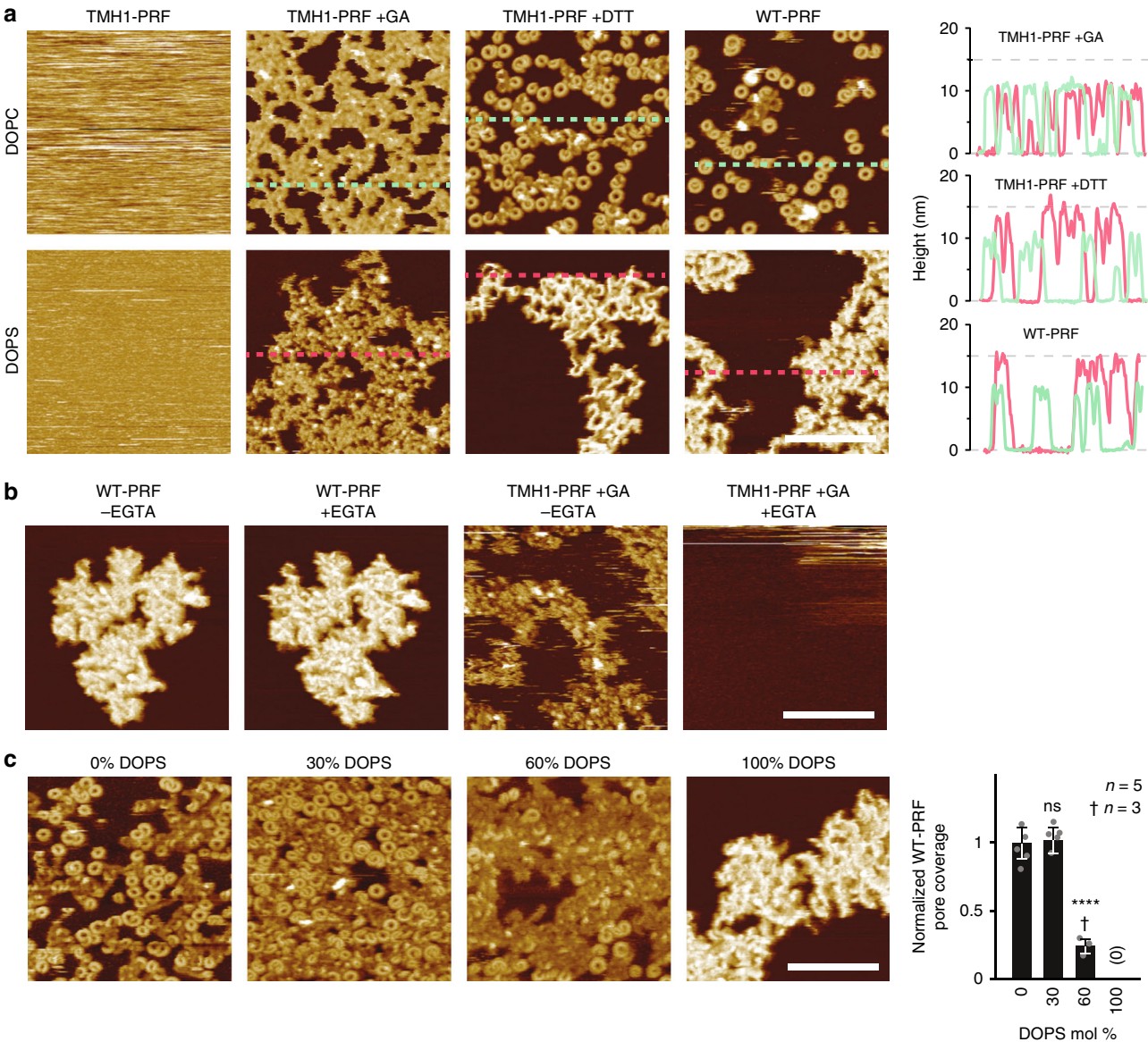

**Fig. 5** Perforin forms non-porating clusters on phosphatidylserine membranes. **a** AFM images of DOPC and DOPS supported lipid bilayers after incubation with TMH1-PRF without or with glutaraldehyde (+GA) fixation, with unlocked (+DTT) TMH1-PRF, or with WT-PRF. Height profiles were plotted as acquired along the dashed lines in the AFM image. On DOPC, prepore locked TMH1-PRF, and TMH1-PRF +DTT and WT-PRF assemblies all extend 10–11 nm above the membrane surface, in agreement with previous observations[5,6,19]. On DOPS, fixed (and locked) TMH1-PRF assemblies have the same height, but the clusters of unlocked TMH1-PRF (+DTT) and WT-PRF are distinctly taller, about 15 nm above the membrane. **b** AFM images of DOPS supported lipid bilayers incubated with WT-PRF or TMH1-PRF +GA, before and after $Ca^{2+}$ chelation by washing the membrane with 5 mM EGTA (±EGTA). Clusters formed by WT-PRF are not removed or visibly affected by washing the membrane with EGTA. In contrast, cross-linked plaques of TMH1-PRF +GA are removed after the EGTA wash. This is consistent with failure of TMH1-PRF to insert into the membrane[6] and suggests that WT-PRF partially inserts into the DOPS membrane. All samples were incubated at 37 °C and imaged at room temperature. Colour (height) scale for all panels as in Fig. 3b. **c** AFM images of supported DOPC bilayers mixed with 0–100% DOPS, after incubation with 150 nM WT-PRF. The images show decreased formation of arc- and ring-shaped perforin pores upon increased DOPS content, and an increased amount of clustering. Bar graph on the right shows quantification (see Methods; mean ± standard deviation) of pore formation on DOPS containing membranes: for 60% DOPS, there is a significant decrease in the number of pores, and at 100% DOPS there are no unambiguous pore features found. Statistical significance was assessed using ANOVA with Dunnett's post-hoc analysis, where 'ns'–not significant, ****$p < 0.001$. Scale bars, 200 nm.

example, killer cells would have to be raised in at least equal numbers to virus-infected cells to ensure clearance of any infection[39,40]—this would be a particular challenge in major organs such as the liver, where many trillions of parenchymal cells are typically infected in a very short time-frame. In addition, if every CTL was to die following interaction with a target cell, this would preclude any antigen-experienced CTL from differentiating into a memory cell. Clearly, such a scenario would have

dire repercussions for the immune system's ability to efficiently eliminate serious pathogens, while compromising antigen recall responses and immune homoeostasis. It is therefore not surprising that mammals have evolved mechanisms to ensure that perforin's deleterious effects on the killer cell are carefully controlled, just as there are multiple mechanisms in place to prevent the premature activation of perforin during its biosynthesis, processing, storage and release[41–43].

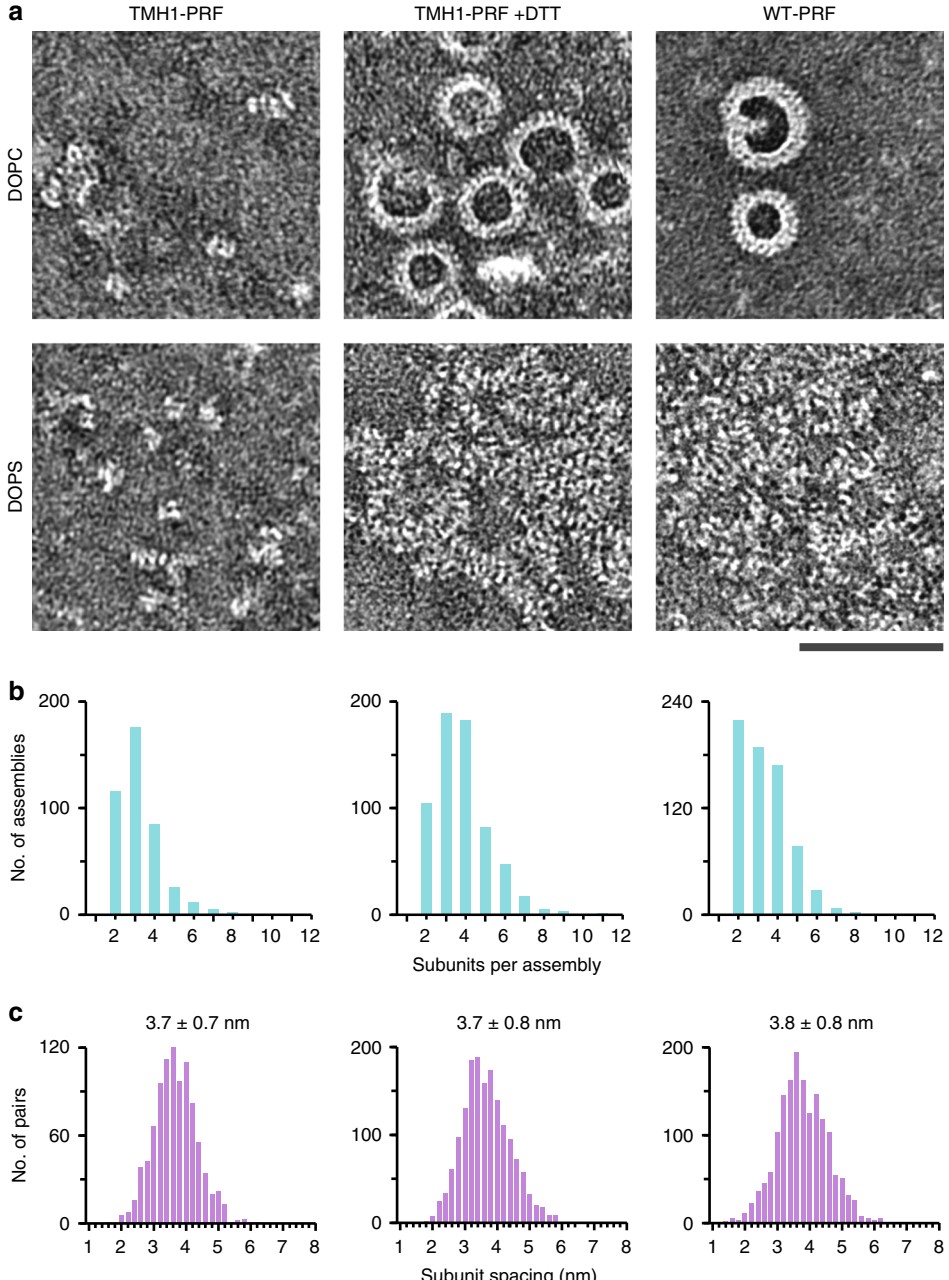

**Fig. 6** On DOPS, perforin assemblies retain properties found in prepores. **a** EM images of DOPC and DOPS monolayers after incubation with TMH1-PRF (−DTT), unlocked TMH1-PRF +DTT, or with WT-PRF, at subunit resolution. On DOPC, TMH1-PRF prepores are visible as short, loose assemblies, which grow into arc- and ring-shaped pores after unlocking (TMH1-PRF +DTT) or when using WT-PRF, accompanied by a tighter subunit packing, and fully consistent with previous observations[6]. On DOPS, initially short assemblies formed by TMH1-PRF appear similar to prepores formed on DOPC. After unlocking with DTT or when incubating with WT-PRF, the clusters retain signatures of short assemblies, and lack the tight packing observed for perforin pores. Scale bar, 50 nm. **b** Length distributions of assemblies as found in isolated assemblies and clusters on DOPS that show subunit repetitions (see Methods). **c** Average subunit spacing (± standard deviation) on DOPS; these results are comparable with previously published data on perforin prepore (TMH1-PRF) assemblies formed on PC-rich membranes, with sizes ranging between 2 and 10 subunits and a subunit spacing of 3.86 ± 0.13 nm; on such PC rich membranes, pore assemblies are significantly larger and have a tighter subunit spacing, 2.55 ± 0.09 nm[6]. Source data for **b** and **c** are provided as a Source Data file.

Here, we have demonstrated that some physiologically relevant types of lipid membrane composition are far less susceptible to perforin binding and/or poration than others; and that CTLs use these more resistant lipid formulations in a two-layered self-defence mechanism to maximize protection. In particular, the lymphocytes become refractory to secreted killer proteins due to increased plasma membrane lipid order, thus reducing perforin binding, and the exposure of negative charge on the membrane surface via PS, thus inactivating residual perforin within the immune synapse. These self-protection mechanisms provide each CTL with the ability to kill many successive targets, which is essential for timely viral or cancer cell clearance and for host immune homoeostasis. This represents a hitherto unsuspected role for lymphocyte lipid organization, going beyond previously demonstrated effects on immune signalling[44]. Finally, our findings suggest a lipid-based

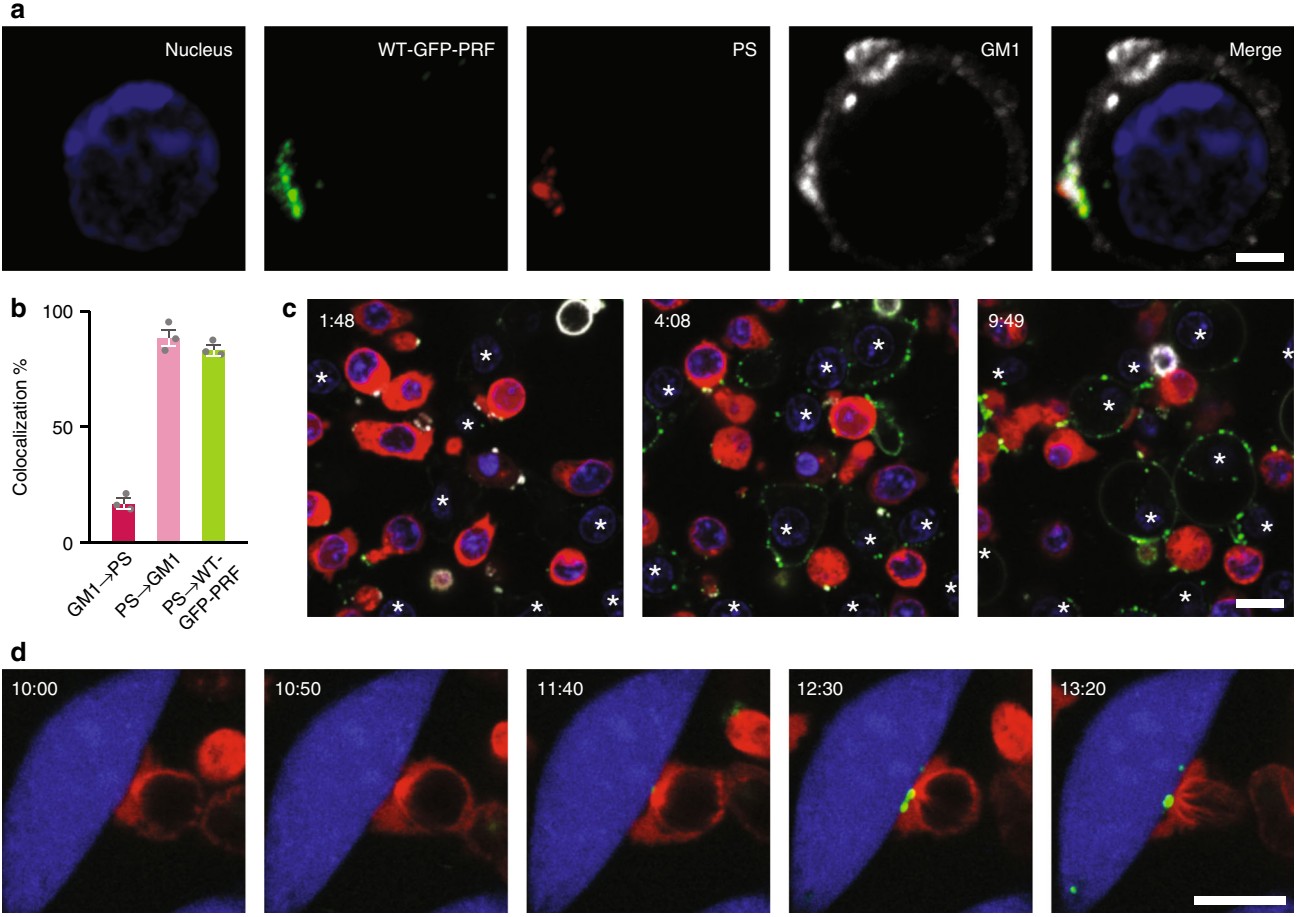

**Fig. 7** Perforin binding to phosphatidylserine on the CTL plasma membrane. **a** Colocalization of GM1 (detected by cholera toxin subunit B-Alexa Fluor 647), externalized PS (detected by annexin V-Alexa 568) and recombinant WT-GFP-PRF (signal enhanced by GFP-TAG polyclonal antibody-Alexa 488) on a CTL. The nucleus (blue) is detected by Hoechst 34580. **b** Quantitative analysis (see Methods) of **a** shows low colocalization levels of GM1 with PS (17%) but high colocalization of PS with GM1 (88%) and PS with WT-GFP-PRF (83%). This indicates that the vast majority of PS is contained within lipid raft areas (whereas there are many lipid raft areas which do not contain PS) and more than 80% of this PS binds WT-GFP-PRF. Sixty-two cells were analysed, and the mean ± s.e.m. of three independent experiments is shown for each condition. Source data are provided as a Source Data file. **c** Montage of time-lapse confocal microscopy (Supplementary Video 1). WT-GFP-PRF (green) was added at 1:17 min to a mixture of CTLs transduced with cherry-tubulin (red) and EL4 cells (white asterisks). Hoechst 33342 staining (blue) shows the nucleus of both cell types. Annexin V-Alexa 647 (white) was maintained in the culture medium throughout the assay; it binds to PS exposed on the CTLs and to PS on and within EL4 cells that are exposed to cytotoxic levels of perforin and that lose membrane integrity. EL4 cells are seen to bind WT-GFP-PRF uniformly before gradually becoming annexin V positive, whilst perforin on the CTL membrane localizes precisely (and almost immediately) to regions of exposed PS, without lysing the CTLs (Supplementary Video 2). **d** Montage of time-lapse confocal microscopy (Supplementary Video 3) reveals a bright punctate region of PS (detected by annexin V-Alexa 488 maintained during assay; shown in green) exposed on the CTL membrane during synapse formation. Cell Trace Violet labelled target cells (MC57, shown in blue) were labelled with SIINFEKL and synapse formation is confirmed by the docking of the centrosome (detected by cherry-tubulin expressed in the $Prf1^{-/-}$ CTL; shown in red) at the point of contact between CTL and target. Timestamps in **c** and **d** are in minutes. Scale bars: **a**, 2 μm; **c**, **d**, 10 μm.

mechanism through which transformed cells might achieve resistance to CTL attack, e.g. in the context of cancer immunotherapies.

## Methods

**Lipid vesicle and AFM sample preparation**. 1,2-dioleoyl-*sn*-glycero-3-phos-phocholine (DOPC), 1,2-dioleoyl-*sn*-glycero-3-phospho-L-serine (DOPS), DOPG, egg SM, N-oleoyl-D-erythro-sphingosylphosphorylcholine (18:1 SM), cholesterol, cholesterol 3-sulphate (CS), and 7KC were purchased from Avanti Polar Lipids (Alabaster, AL, USA) as powder or dissolved in chloroform or in case of CS in a chloroform/methanol/water mixture (20:9:1 volume ratio). The lipids were mixed in the desired molar ratio, and small unilamellar vesicles were prepared using the lipid extrusion method[6,45]. Indicated lipid ratios and percentages refer to molar values and are given with ±5% confidence intervals.

A total of 4 μL of the small unilamellar vesicles were injected onto a freshly cleaved mica surface (Agar Scientific, Essex, UK) in the presence of 80 μL of 20 mM HEPES, 150 mM NaCl, 25 mM $Mg^{2+}$, 5 mM $Ca^{2+}$, pH 7.4, adsorption buffer. To form pure DOPG bilayers, an adsorption buffer containing 10 mM $Ca^{2+}$ and no $Mg^{2+}$ was used. A 30 min incubation above the main transition temperature of the

lipids allowed the vesicles to rupture and adsorb onto the mica surface, yielding an extended lipid bilayer film. Excess vesicles were removed by washing the samples at least nine times with 80 μL of the adsorption buffer. Supported lipid bilayers containing DOPS (Figs. 5, 6), DOPG, or CS (Supplementary Fig. 8) were washed with 20 mM HEPES, 150 mM NaCl, 5 mM $Ca^{2+}$, pH 7.4, instead of the adsorption buffer to remove $Mg^{2+}$ or excess $Ca^{2+}$ ions, as they would interfere with perforin binding to the (negatively charged) membranes.

Perforin was diluted up to ca. tenfold in 20 mM HEPES, 150 mM NaCl and injected onto supported lipid bilayers in the adsorption buffer, leading to a final protein concentration of about 150 nM, and incubated for 5 min at 37 °C. The AFM samples used in Fig. 3 were incubated for a reduced duration (2 min instead of 5 min) and subsequently washed six times with the adsorption buffer to counteract excessive aggregation on some of the lipid mixtures under study.

For unlocking TMH1 perforin and TMH1-GFP perforin after their binding to the membrane (Figs. 5, 6; Supplementary Fig. 5), the mutant proteins were incubated with 2 mM DTT (Sigma-Aldrich) for 10 min at 37 °C.

Mobile, membrane-bound TMH1-PRF in Fig. 5 was fixed by addition of 0.04% glutaraldehyde (TAAB Laboratories, Reading, UK), labelled +GA in the figures) and 10 min incubation at room temperature. The samples were next washed with 20 mM HEPES, 150 mM NaCl, 5 mM $Ca^{2+}$, pH 7.4, prior to AFM imaging.

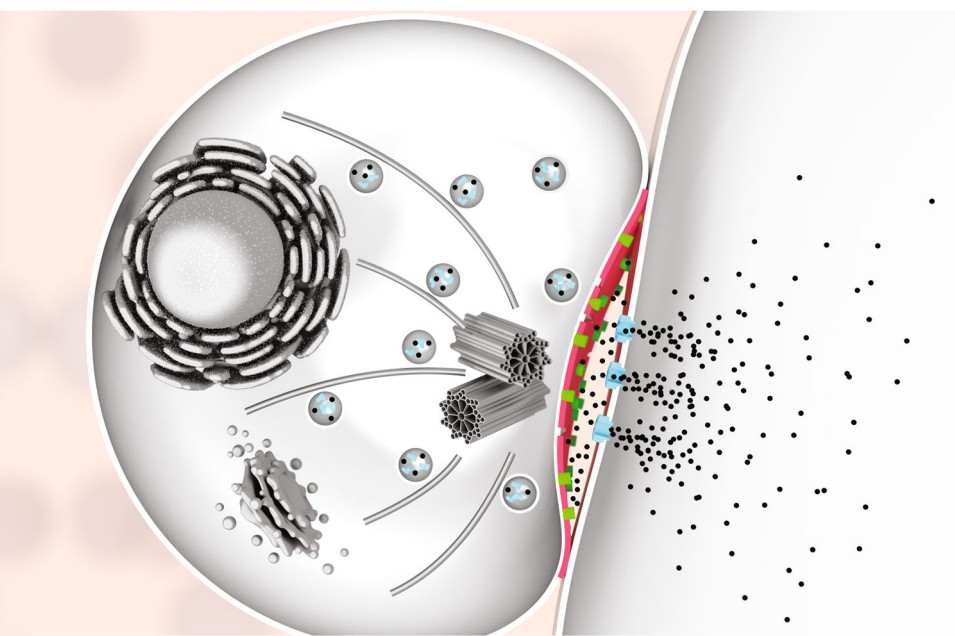

**Fig. 8** Mechanism of cytotoxic lymphocyte protection from secreted perforin. Illustration of a cytotoxic lymphocyte forming an immune synapse (shown are the microtubule-organizing centre and cytotoxic granule polarization), with granzymes (black dots) entering the target cells through perforin pores (light blue) in the target cell plasma membrane. High lipid order domains (red) and exposed phosphatidylserine (green) at the immunological synapse protect the lymphocyte against the perforin it secretes.

**AFM imaging and data processing**. AFM images were recorded by force-distance curve-based imaging (PeakForce Tapping) on a MultiMode 8 system (Bruker, Santa Barbara, CA, USA). PeakForce Tapping was performed at 2 kHz and a maximum tip-sample separation between 5 and 20 nm. Images were typically recorded at ca. 6 min/frame on an E scanner (Bruker, Santa Barbara, CA, USA) with temperature control. MSNL-E and -F probes (Bruker, Santa Barbara, CA, USA) were used at forces between 50 and 100 pN.

Raw AFM data were processed in NanoScope Analysis software version 1.80 (Bruker, Santa Barbara, CA, USA). AFM images were flattened with respect to the lipid surface, using a height threshold and second-order flattening. Height values indicated in the manuscript are given with ±1 nm confidence intervals, with the uncertainty related to both scanner calibration and the possible sample deformation in the AFM images. Values for perforin coverage (Fig. 3) were estimated by considering the number of pixels above a height threshold, located 6–8 nm above the membrane surface and adjusted to counteract broadening effects of differently shaped AFM tips. The values for perforin coverage obtained by this method are either given as area percentage, or as values between 0 and 1 when normalized with respect to a 100% DOPC reference. In the case of Fig. 5c and Supplementary Fig. 8, for which we recorded data at a higher pixel resolution, the perforin coverage was determined by tracing pore shapes with 3dmod 4.9.4 (BL3DEMC & Regents of the University of Colorado,[46]). The traces were normalized with respect to a reference coverage as measured on pure DOPC membranes, resulting in values of coverage between 0 and 1.

**Electron microscopy**. Negatively stained samples of perforin on pure DOPC and DOPS monolayers were prepared in polytetrafluoroethylene (PTFE) troughs[6,47] containing 35 μL of buffer (20 mM HEPES, 150 mM NaCl, 5 mM Ca$^{2+}$, pH 7.4) containing ca. 1.2 nM WT- or TMH1-PRF. In brief, 0.3 μL of 1 mg/mL lipid in chloroform was placed at the air/buffer interface to form a lipid monolayer. After 1 min, a 200-mesh carbon-film-coated gold grid (Agar Scientific, Essex, UK) was placed on the lipid monolayer and incubated for 15 min at 37 °C (for TMH1-PRF, 2 mM DTT was added after 5 min). The samples were subsequently washed and stained with 2% w/w uranyl acetate.

Electron micrographs were recorded on a 4kx4k Ultrascan 4000 CCD camera (Gatan, Pleasenton, CA, USA), using a Tecnai T12 microscope (FEI Company, Hillsboro, OR, USA) operating at an accelerating voltage of 120 kV, a magnification of 67k (1.64 Å pixel size), and an underfocus of 0.5–1 μm. Final adjustments on the raw images were done in Fiji/ImageJ 1.50e[48]. Raw images were processed with a Gaussian bandpass filter with a range of 4–300 pixels and binned by a factor of two to reduce noise, yielding a 2048 × 2048 pixel image. The contrast was adjusted with the Auto setting in the Fiji/ImageJ software.

In EM data with well-resolved assemblies and assembly subunits in perforin clusters on DOPS (Fig. 6), histograms of assembly sizes and subunit distances were obtained from a custom interface written in Matlab R2016A (Mathworks, Cambridge, UK). Within the interface, perforin subunits were located and marked by hand within each assembly. In the EM images recorded on DOPS, perforin assemblies appeared sufficiently separated to distinguish them from one another, and their subunits were sufficiently clear to allow such manual tracing.

**Cell culture**. Murine cell lines EL4 (ATCC TIB-39), P815 (ATCC TIB-64) and MC57 (ATCC CRL-2295) were maintained in SAFC DMEM (Sigma-Aldrich, Missouri, USA) media supplemented with 10% FCS (Gibco, Loughborough, UK), 15 mM HEPES (Merck, Missouri, USA), 44 mM NaHCO$_3$ (Merck, Missouri, USA) and 2 mM Glutamax (Gibco, Loughborough, UK) at 37 °C in 10% CO$_2$. Primary murine CD8$^+$ T cells (CTLs) and purified natural killer cells were generated from BL/6 OTI transgenic mice or BL/6 mice, respectively, and maintained as described previously[49] (approved by the Peter MacCallum Cancer Centre Animal Ethics Committee).

Wild-type perforin (WT-PRF), perforin-GFP fusion protein (WT-GFP-PRF), TMH1 perforin mutant (TMH1-PRF) and its GFP fusion protein (TMH1-GFP-PRF) were expressed and purified using baculovirus expression system[6,15].

**Sorting/surface staining analysis of transduced cells**. To obtain a truncated form of CD107a (LAMP-1) that is retained on the cell membrane[13], the sequence was cloned into the MSCV-IRES-GFP retroviral vector and transduced into EL4 and $Prf1^{-/-}$ OTI T cells[6]. After sorting for equal protein expression levels of the truncated CD107a construct and of an empty vector control (via GFP fluorescence), cells were stained (on the day of the $^{51}$Cr release assay) with anti-CD107a-phycoerythrin (PE) antibody (eBioscience, California, USA) to assess surface levels of CD107a (Supplementary Fig. 3a). Cherry-tubulin fusion[50] was cloned into an MSCV vector, naive CTLs transduced and Cherry-positive cells were sorted 3 days later and used in experiments shown in Fig. 7c, d, Supplementary Fig. 9 and Supplementary Videos 1–3.

**Cytotoxicity assay**. For $^{51}$Cr release assays[51] (Fig. 1a, Supplementary Figs. 1a, 2b, c), $2 \times 10^6$ target cells were incubated with 200 μCi of $^{51}$Cr (sodium chromate) in 200 μL of complete DMEM media for 1 h at 37 °C. Where required for antigen-dependent CTL killing assay (Supplementary Fig. 3c), 1 μM SIINFEKL peptide (GenScript, New Jersey, USA) was included in this incubation step. After 1 h, the cells were washed three times with complete DMEM and either incubated with OTI T cells at the desired effector/target ratio for 4 h, or mixed with various amounts of recombinant perforin and incubated for 1 h; these assays were conducted in 96-well plates in either 200 μL (OTI T cell assays) or 100 μL reactions (recombinant perforin assays). The plates were then centrifuged, supernatant collected, and its radioactivity assessed using a 1470 Wizard Automatic Gamma Counter (Wallac, Turku, Finland). Percentage specific $^{51}$Cr release was calculated as $[(^{51}Cr_{assay} - ^{51}Cr_{spontaneous})/(^{51}Cr_{total} - ^{51}Cr_{spontaneous}) \times 100]$; $^{51}Cr_{total}$ was the level of radioactivity in target cells lysed with 1% Triton X-100, and $^{51}Cr_{spontaneous}$ was the level

of radioactivity released by target cells incubated in the media in the absence of CTL or recombinant perforin for 4 h or 1 h, respectively.

**Perforin binding assays accessed via flow cytometry.** For the flow cytometry assays of perforin binding (Figs. 1b, 2, 4d), cells were washed three times in DMEM containing 0.1% BSA (Roche Diagnostics, Mannheim, Germany) and resuspended at $10^6$ cells/mL. EL-4 (not pulsed with the SIINFEKL antigen) and CTLs were then mixed 1:1 to remain at a final concentration of $10^6$ cells/mL. WT-GFP-PRF or TMH1-GFP-PRF was added to the mixture, and cells were incubated at 4 °C or 37 °C for 30 min. Unbound perforin was removed by washing the cells in 0.1% BSA DMEM, cells were stained with anti-CD8 APC (eBioscience, California, USA) and analysed using a Fortessa X20 flow cytometer (BD Biosciences, New Jersey, USA). To demonstrate $Ca^{2+}$-specific perforin binding, cells were treated with 2 mM EGTA prior to staining with anti-CD8 APC.

**Surface staining for GM1 analysis.** Cells were washed three times in complete DMEM and resuspended at $10^6$ cells/mL. EL-4 (not pulsed with the SIINFEKL antigen) and CTLs were then mixed 1:1 to remain at a final concentration of $10^6$ cells/mL. Cells were stained with anti-CD8 PE antibody (eBioscience, California, USA) and CTxB-Alexa Fluor 647 (Molecular Probes, Oregon, USA) and analysed using a Fortessa X20 flow cytometer (BD Biosciences, New Jersey, USA) (Supplementary Fig. 7).

**Unlocking of TMH1-GFP-PRF on cells.** TMH1-GFP-PRF was added to $^{51}$Cr-labelled EL4 cells resuspended in DMEM supplemented with 0.1% BSA at 37 °C. After 30 min, cells were washed with serum-free media, and 0.75 mM DTT was added to unlock the protein. After 5 min, DTT was quenched by addition of 0.1% BSA, and cells were incubated for a further 2 h at 37 °C (Supplementary Fig. 1a).

**Calcium flux assay.** CTLs and EL4 cells were labelled separately with a ratiometric (400 nm/475 nm) calcium fluorophore Indo-1AM (Invitrogen, California, USA)[8,15] and treated with varying amounts of WT-GFP-PRF to determine amounts for which both cell types stained with a similar level of GFP (Supplementary Fig. 1b). Time-course flow cytometry was performed[15] following WT-GFP-PRF addition (Fig. 1c). Cells treated with 1 μg/mL ionomycin (Sigma-Aldrich, Missouri, USA) were used as controls (Supplementary Fig. 1c).

**Modification of cellular cholesterol using 7KC.** Cholesterol or 7KC (both Sigma Aldrich, Missouri, USA) were dissolved in 100% ethanol at 15 mg/mL and mixed at different ratios. Cholesterol/7KC mixtures (as well as 100% cholesterol designated as 0% 7KC in Fig. 4, or 100% 7KC) were added drop-wise over a period of 30 min, to a solution of 50 mg/mL methyl-ß-cyclodextran (MßCD, Sigma Aldrich, Missouri, USA) in PBS, which was heated to 80 °C, to achieve the final sterol stock concentration of 1.5 mg/mL. Cells were washed three times in 0.1% BSA RPMI-1640 (Gibco, Paisley, UK) before being resuspended at $0.5 \times 10^6$/mL. Up to 2.25 μL of cholesterol/7KC stock solutions in MßCD were then added to 1 mL of cells; the cells were incubated at 37 °C for 30 min, then washed three times in 0.1% BSA DMEM, resuspended at $10^6$/mL and immediately used for either perforin binding or lysis experiments, or laurdan microscopy (Fig. 4).

**Assessment of membrane order via laurdan microscopy.** Cells were incubated in 0.1% BSA DMEM supplemented with 5 μM laurdan (Molecular Probes, Oregon, USA) in DMSO for 1 h. Cells were then pelleted and resuspended in 250 μL serum-free DMEM before being plated out in 8-well Nunc Lab-Tek II #1.5H glass-bottom chamber wells (Thermo Fisher Scientific, Massachusetts, USA). After allowing the cells to adhere for 20 min at 37 °C, 50 μL of 0.5% BSA solution was added to maintain cell viability during imaging. Lambda stacks were recorded using a Zeiss Elyra PS.1 microscope with a Tokai Hit stage/objective heater attached, and 5% CO2/humidity maintained (Zeiss, Oberkochen, Germany). An optical zoom of 2 was applied to a Plan-Apochromat 63×/1.4 Oil DIC lens (Zeiss, Oberkochen, Germany) and images were obtained every 8.9 nm from 410 nm to 695 nm using a line average of 16. Recorded stacks were exported to the Spectral Imaging Toolbox in Matlab[52], where the data were segmented to isolate the laurdan signal from the plasma membranes of individual cells and the generalized polarization (GP) of the plasma membrane calculated (Fig. 4a). Images of a reference solution (laurdan in DMSO) were obtained with the same microscope settings as used for the imaging of cells, and a reference value ($GP_{ref}$) of 0.207 was used for laurdan[26]. For details of GP calculation using the Spectral Imaging Toolbox, see ref. [52].

**Live cell imaging of annexin V-labelled cells.** For Fig. 7c, Supplementary Fig. 9 and Supplementary Videos 1 and 2, a 1:1 mixture of EL4 and Cherry-tubulin expressing OTI T cells was washed three times in 0.1% BSA DMEM, labelled with 5 μM Hoechst 33342 (Life Technologies, California, USA), washed three times and then resuspended in a 1:50 stock dilution of annexin V-Alexa 647 in 0.1% BSA DMEM. These cells were then plated into 8-well Nunc Lab-Tek II #1.5H glass-bottom chamber wells (Thermo Fisher Scientific, Massachusetts, USA). WT-GFP-PRF was added during imaging. Cells were imaged using a Zeiss Elyra PS.1 microscope with a Tokai Hit stage/objective heater attached, and 5% CO2/humidity

maintained (Zeiss, Oberkochen, Germany). Imaging was performed using a Plan-Apochromat 63×/1.4 Oil DIC lens (Zeiss, Oberkochen, Germany) using an optical zoom of 1 and a line average of 4.

For live-cell imaging of PS exposure at the immunological synapse (as detected by annexin V labelling) (Fig. 7d, Supplementary Video 3), MC57 target cells were trypsinized and washed. A total of $10^6$ cells were then resuspended in 500 μL of 5 μM Cell Trace Violet solution (Invitrogen, California, United States) in PBS for 20 min at 37 °C; 10 mL of complete medium was then added to these cells to quench any unbound dye. Cells were pelleted, resuspended in complete DMEM media, counted and plated at 30,000 cells ($10^6$ cells/mL) per well of an 8-well ibidi-treat imaging chamber (Ibidi, Martinsried, Germany), 1 day before imaging. Approximately 4 h before imaging, cells were labelled with 1 μM SIINFEKL (GenScript, New Jersey, USA) at 37 °C for 1 h, before being washed three times in complete DMEM, and allowed to rest. Cherry tubulin transduced OTI T cells from $Prf1^{-/-}$ mice were then resuspended in 300 μL of a 1:50 stock dilution of annexin V-Alexa Fluor 488 (Invitrogen, California, USA) in complete DMEM before being added to the microscopy chamber. All medium was removed from the MC57 cells, such that both the MC57 and OTI T cells were imaged in the presence of a 1:50 stock dilution of annexin V-Alexa Fluor 488. Cells were imaged using a Zeiss Elyra PS.1 microscope with a Tokai Hit stage/objective heater attached, and 5% CO2/humidity maintained (Zeiss, Oberkochen, Germany). Imaging was performed using using a C-Apochromat 63×/1.2W Korr UV-VIS-IR lens (Zeiss, Oberkochen, Germany) using an optical zoom of 1 and a line average of 8.

**Fixed cell samples for colocalization analysis.** Hydrophobic barriers were drawn on #1.5H coverslips (Menzel Glaser, Leicestershire, UK) using a mini-PAP PEN (Life Technologies, California, USA) and the resulting wells coated with 0.1 mg/mL poly-L-lysine (Sigma Aldrich, Missouri, USA) and dried for >2 h. OTI T cells were pre-labelled with annexin V-Alexa Fluor 568 (Invitrogen, California, USA) for 10 min at room temperature in 0.1% BSA DMEM. After washing three times in 0.1% BSA DMEM, cells were then treated with WT-GFP-PRF for 30 min at 37 °C, washed three times and added to the wells. The cells were allowed to adhere for 10 min at 37 °C, and the wells washed gently with Hanks Balanced Salt Solution (HBSS, Sigma Aldrich, Missouri, USA) containing 2.5 mM CaCl2 and 1 mM MgCl2 (both from Sigma–Aldrich, Missouri, USA) to remove any unbound cells. Cells were fixed with 4% EM Grade paraformaldehyde (Electron Microscopy Sciences, Pennsylvania, USA) (in HBSS, with 2.5 mM CaCl2) for 30 min at room temperature and washed with 0.1 M lysine (Sigma-Aldrich, Missouri, USA) in HBSS with 2.5 mM CaCl2. After 30 min, lysine was removed, and samples washed four times with HBSS with 2.5 mM CaCl2. HBSS containing 2.5 mM CaCl2 and 2% BSA was then added to the wells and incubated for 1 h at room temperature (with gentle shaking) to block non-specific antibody binding. A 1:100 stock dilution of GFP-tag polyclonal antibody-Alexa Fluor 488 (Invitrogen, California, USA) and 2 μg/mL CTxB-Alexa Fluor 647 (Molecular Probes, Oregon, USA) in HBSS containing 2.5 mM CaCl2 and 2% BSA was then added to wells and incubated overnight at 4 °C. The cells were washed four times and incubated with 5 μg/mL Hoechst 34580 (Life Technologies, California, USA) for 30 min at room temperature. Cells were subsequently washed four times and allowed to dry before being mounted onto slides using Vectashield mounting medium (H1000, Vector Laboratories, California, USA) and sealed.

Fixed samples (Fig. 7a, b) were imaged using a Zeiss Elyra PS.1 microscope, using sequential imaging in line-scan mode. A Plan-Apochromat 63×/1.4 Oil DIC lens (Zeiss, Oberkochen, Germany) was used with zoom set to 2 and a pixel size of 70 nm. Frame size was set to 1024 × 1024 and samples were imaged using a line average of 8 and a z-stack interval of 200 nm. Background fluorescence levels for each channel were obtained using negative control images (primary antibody omitted) of the same acquisition setting and used to set thresholds for colocalization analysis. A channel alignment slide containing 200 nm multispec beads (Zeiss, Oberkochen, Germany) was imaged and used as a reference to correct for chromatic aberration. An ImageJ macro was written to segment individual cells from a field of view and to create mask channels in Fiji/ImageJ.

Colocalization analysis was performed on individual cells using Imaris 8.4.2 (Bitplane, Belfast, UK). The analysis was performed in the mask channels created previously. The threshold of each channel was set as the maximum intensity value of the negative control. The percentage of colocalization of a channel (Fig. 7b) was calculated as the sum of colocalized intensity above the threshold divided by the sum of total intensity above the threshold:

$$Colocalization = \frac{\sum I_{above\ threshold}^{colocalized}}{\sum I_{above\ threshold}^{total}} \times 100. \quad (1)$$

**Reporting Summary.** Further information on research design is available in the Nature Research Reporting Summary linked to this article.

## Data availability
Data supporting the findings of this manuscript are available from the corresponding authors upon reasonable request. A reporting summary for this Article is available as a Supplementary Information file. The source data underlying Figs. 1, 2c, 3a, 4, 5b, 6b, c, 7b, and Supplementary Figs. 1, 3b, c, 7, and 8b are provided as a Source Data file.

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

## Acknowledgements

We thank Richard Thorogate and Shu Chen for technical support; Natalya Lukoyanova for advice on electron microscopy experiments; Helen Saibil, Ricardo Henriques and Dylan Owen for discussions and comments on the manuscript. We also thank the following Peter MacCallum Cancer Centre core facilities for technical support: Ralph Rossi and the Flow Cytometry facility, Sarah Ellis and the Centre for Advanced Histology and Microscopy. This work has been funded by the NHMRC Project Grant (1128587), the NHMRC Fellowship (1059126), the BBSRC (BB/J005932/1, BB/J006254/1 and BB/N015487/1); the EPSRC (EP/M028100/1); and the Sackler Foundation. This research was supported by the Australian Cancer Research Foundation (for the Peter MacCallum Cancer Centre core facilities).

## Author contributions

J.A.R.-S. designed, performed and analysed cell-based and microscopy assays and co-wrote the paper. A.W.H. designed, performed and analysed AFM and EM experiments

and co-wrote the paper. T.N. performed and analysed cell-based assays. J.A.L. performed cell-based assays. H.-J.C. designed and analysed the microscopy-based assays. S.V. and A. C. expressed and purified perforin. J.A.T. conceived the study and co-wrote the paper. B. W.H. and I.V. contributed equally, co-led this work, and jointly conceived and supervised the study, designed the experiments, analysed the data and wrote the paper. All authors read and commented on the paper.

## Competing interests

The authors declare no competing interests.
