## [Peer Review File · Nature Communications]

Reviewers' Comments:

Reviewer #1:

Remarks to the Author:

Rudd-Schmidt et al. have provided a substantial amount of data to support their conclusion that CTLs protect themselves from the action of perforin at the immunological synapse by altering the membrane composition. They convincingly demonstrate that increasing the levels of ordered lipids decrease perforin binding to CTL membranes. Furthermore, they show that the perforin that does bind is preferentially bound to PS which appears to prevent its transition to the pore. They also show that lamp-1, previously suggested by Cohnen et al to protect the CTL from the actions of perforin at the immunological synapse, does not appear to play a role in protection to the action of perforin. The experiments are generally well controlled and extensive, and the work with artificial membranes is complemented nicely with the cell work. The studies provide a significant advancement into our understanding about how CTLs protect themselves against the actions of perforin, so that they can continue to attack and eliminate cells, which have been compromised by virus infection, tumorigenesis, etc. The Discussion is, however, a bit terse.

1. I don't necessarily disbelieve the data in Fig. 2b but the authors indicate that the dashed lines are the phase boundaries (presumably between liquid ordered and disordered). How did they detect the two phases other than binding of the perforin. They can't really use perforin to define phase boundaries, as it is the experiment, not the control. There does not appear to be an independent set of probes for the liquid ordered and disorder phases and their phase boundaries.
2. In figure 6 is the lysis of the EL4 cells is a low probability event since perforin does not directly lyse the cells, rather they eventually rupture from the action of the granzymes? The reason this is asked is that only 1 or 2 EL4 cells appear to be stained for PS so it might be worth a brief explanation that lysis of those cells is not going to occur rapidly.
3. The authors may want to discuss/speculate on the PS bound perforin: is this bound in such a way as to be permanently inactive? If so is there any evidence that it is endocytosed and eliminated? This would be a necessary action to restore the ability of the immunological synapse to function again.
4. The authors state, in reference to Fig. 5, starting on line 178 that "Taken together, these observations on PS are consistent with perforin assembly that is trapped in a dysfunctional (dead-end) state upon its initial insertion into the membrane, failing to complete the prepore-to-pore transition.". I think that the phrase "initial insertion into the membrane" is misleading, as it implies a dysfunctional insertion of the β -barrel pore. It appears that perforin "binds" to the PS membrane but cannot convert to a closed ring. Therefore, binding to the PS appears to prevent the monomers from undergoing the conformational changes necessary to the proper interaction of monomers, which changes the geometry of the interaction so that the monomers are stuck a more open, curvilinear configuration and cannot convert to a circular pore.
5. In Fig. 4 the authors need to denote the colors in the figure legend for panel 4a, right-most panel. Also, the increased height of the unlocked prepores of 5 nm is fairly substantial, it is 40% of the height of the perforin molecule. Do the authors have any explanation for this quite dramatic increase in height? It is hard to envision how this happens based on what we know about the mechanism.

Reviewer #2:

Remarks to the Author:

Cytotoxic cells form an immune synapse with their target cells. Into the confined space of the immune synapse, the cytotoxic cells then release perforin that forms pores in the target cell's plasma membrane. Now, apoptotic serine proteases (granzymes) enter the target cell and initiate apoptosis. The cytotoxic cell is not affected, despite being exposed to its own secretion into the cleft of the immunological cleft. The manuscript addresses, how the cytotoxic cells protect

themselves against their own toxic secretions.

The authors argue that the protection is afforded by prevention of pore formation by perforin. They claim two mechanisms. Firstly, they show that perforin requires a disordered lipid phase to form pores in vitro, in AFM. They show that a reduction in membrane order of cytotoxic cells by loading with a cholesterol analogue that does not support lipid raft formation highly increases lysis of these cells by perforin. They show a moderate increase in this respect for target cells that are already readily lysed.

They further show a role of phosphatidylserine. They show in vitro in both TEM and AFM that PS disrupts pore formation and leads to clusters. They show that perforin in CTLs co-localizes with PS. They claim that in the immunological synapse, PS is introduced to the outer leaflet of the CTL membrane and prevents pore formation.

Assessment:

In my opinion, this is a well executed study that demonstrates the possibility that lipid organization is responsible for the protection of cytotoxic cells from their own toxic secretion. Cell biological- and biophysical methods are well designed and executed. Data analysis is adequate. Overall, I am excited to see attention paid on the role of lipids for this important biological function and I am excited about this paper overall.

Strength (this is what I believe the authors have proven):

- that the perforin binds much less to CTLs than target cells and that these cells are therefore protected.
- that disruption of lipid rafts reverses this protection.
- that perforin forms pores only in the lipid disordered phase of phase separated membranes.
- that PS or other negatively charged membrane lipids sequester perforin into dysfunctional aggregates rather than pores.

Points where additional experiments and interpretations are warranted:

In my assessment, the conclusions that CTL are protected by a high-order lipid membrane and that PS is externalized by CTLs as an additional protective mechanism are not yet fully supported by the evidence. The authors have proven that this is a possibility.

Specifically, the authors do not show that the overall lipid order in CTLs is greater than in target cells. This should be easy for them, using the lipid order sensor laurdan. This is significant for this manuscript, in my opinion. A great number of membrane functions is associated with lipid rafts. Most membrane proteins are either raft- or non-raft proteins and may loose- or change their function if the membrane order is altered. Disrupting rafts changes or abolishes protein-dependent and -independent functions alike.

Seeing that pore formation is likely a simple self-assembly process and is apparently not dependent on other proteins I still believe that the authors have a strong point, particularly when also considering their work on model membranes. It is however important to strengthen this point by comparing CTLs to target cells. As well, the GP changes quite moderately: does the change support the dramatic effect the authors have observed in terms of pore formation? Moreover, does the moderately lower GP after treatment in CTLs equate the status in target cells pre-treatment? Unless these additional experiments provide a clear-cut prove, I would put the lipid-order based interpretation of the results forth as a strong possibility rather than a certainty.

Regarding the role of PS: Does PS co-localize with perforin in target cells as well (i.e. is this a consequence of membrane association of perforin)? This should be established as a control. The video S3 shows PS in the contact zone of the two cells. Whether PS is on one or the other cell or on both is not resolved. Therefore, the evidence is not strong enough to proclaim this a "mechanism", in my opinion and may be better described as a possibility. Further experiments may clarify this point.

Overall, this is an exciting manuscript.

Matthias Amrein

Reviewer #3:

Remarks to the Author:

This is a very interesting and highly novel paper that convincingly demonstrates that the biophysical properties of plasma membrane protects cytotoxic T lymphocytes (CTLs) from perforin. The experiments are well performed and I have only minor issues:

- In the abstract, the authors state that the resulting resistance of CTLs to perforin explains the ability to kill target cells in rapid succession. Is the CTL membrane remodeled before and after perforin release or is phosphatidylserine permanently at elevated levels in the outer leaflet of the plasma membrane?
- The membrane order is higher in the periphery of the immunological synapse compared to the center of the synapse (Ref 33 and 34). Would the authors expect more perforin binding to the center of the synapse and have they observed this in CTL synapses?
- In the experiment shown in Figure 4, are the charged lipids required per se for perforin assembly or are the charged lipid required to create a surface potential? In other words, can assembly be blocked by increasing the ionic strength in solution?
- An interesting but optional experiment would be to induce phosphatidylserine to the outer leaflet of the plasma membrane in target cells and examine whether elevated phosphatidylserine levels are sufficient to prevent CTL killing.

I would like to emphasize that all of the proposed experiments are optional (but may improve the manuscript); they are not required for publication.

Rudd-Schmidt, Hodel, et al., “Lipid order and charge protect killer T cells from accidental death”

Response to Reviewers

Reviewer #1 (Remarks to the Author):

Rudd-Schmidt et al. have provided a substantial amount of data to support their conclusion that CTLs protect themselves from the action of perforin at the immunological synapse by altering the membrane composition. They convincingly demonstrate that increasing the levels of ordered lipids decrease perforin binding to CTL membranes. Furthermore, they show that the perforin that does bind is preferentially bound to PS which appears to prevent its transition to the pore. They also show that lamp-1, previously suggested by Cohnen et al to protect the CTL from the actions of perforin at the immunological synapse, does not appear to play a role in protection to the action of perforin. The experiments are generally well controlled and extensive, and the work with artificial membranes is complemented nicely with the cell work. The studies provide a significant advancement into our understanding about how CTLs protect themselves against the actions of perforin, so that they can continue to attack and eliminate cells, which have been compromised by virus infection, tumorigenesis, etc. The Discussion is, however, a bit terse.

Response: We thank the reviewer for this supportive feedback on our manuscript, and here provide a point-by-point response to his/her specific comments below.

1. I don't necessarily disbelieve the data in Fig. 2b but the authors indicate that the dashed lines are the phase boundaries (presumably between liquid ordered and disordered). How did they detect the two phases other than binding of the perforin. They can't really use perforin to define phase boundaries, as it is the experiment, not the control. There does not appear to be an independent set of probes for the liquid ordered and disorder phases and their phase boundaries.

Response: We defined the lipid phase boundaries not based on perforin coverage, but based on the locally measured height of the membrane, as an independent probe to distinguish between more and less ordered lipid domains. The reviewer's comment has made us realise that the colour scale in Fig. 2b did not provide an adequate visual representation of the subnanometre differences in membrane thickness (and thus in measured height) that allowed us to define the membrane phase boundaries as drawn in Fig. 2b; this colour scale was optimised for visualising the perforin pores, which protrude ~10 nm above the membrane. To better highlight how we distinguish between different lipid domains, we have here added alternative representations of the same AFM images, with the colour scale adjusted (now showing perforin pores saturated, white) such that the differences between the two lipid phases are obvious from the measured height (colour) of the membrane surface, see Figure R1. To clarify this in the manuscript, we have enhanced the colour contrast for a part of Fig. 2b(ii), and amended the caption accordingly.

Figure R1: Data shown in the manuscript in Fig. 2b, subfigures ii (here a) and iii (here b) in a different colour scale (indicated on the top right). Different phase domains are visible in the membrane: the thicker (higher) liquid ordered domains and the thinner (lower) liquid disordered domains. The liquid disordered domains are decorated with perforin pores (bright white, i.e. outside the colour scale).

2. In figure 6 is the lysis of the ELA cells is a low probability event since perforin does not directly lyse the cells, rather they eventually rupture from the action of the granzymes? The reason this is asked is that only 1 or 2 ELA cells appear to be stained for PS so it might be worth a brief explanation that lysis of those cells is not going to occur rapidly.

Response: The Reviewer refers to Figure 6c, which was derived from Video S1. There, we only used perforin (no granzyme B was added to the assay). We added a concentration of perforin that would not induce a rapid lysis of cells, as this would preclude us from visualising their interaction with perforin. Unlike in the case of erythrocyte-based lysis assays, where the cells cannot actively repair perforin pores, nucleated

cells employ an exocytic membrane repair response that is capable of protecting cells from an acute perforin lysis (as per Ref 15); however, more cells will die from perforin-induced damage by 40-60 min. As the rate of cell death depends on the concentration of perforin (number of pores formed on the cell membrane), we optimised the assay in order to investigate the interaction between WT-GFP-PRF and EL4/CTLs. We, therefore, have included two videos – Video S1 shows “uniform” perforin binding to EL4 within 10 min without causing overt lysis, and Video S2 shows a long-term effect of the same amount of perforin, with CTLs remaining intact throughout, but EL4 cells becoming “uniformly” double-positive for WT-GFP-PRF and annexin V, indicating cell death.

Following this reviewer comment, we have now adjusted the figure legend of Figure 6c to include the following explanation of why lysis of EL4 cells is not seen immediately: ‘Nucleated cells employ an exocytic membrane repair response that is capable of protecting cells from an acute perforin lysis (Ref 15), meaning EL4 cells do not immediately become annexin V positive, at least not for the perforin concentrations used here. However, more EL4 cells will die from perforin-induced damage by 40-60 min, as highlighted in Video S2.’

3. The authors may want to discuss/speculate on the PS bound perforin: is this bound in such a way as to be permanently inactive? If so, is there any evidence that it is endocytosed and eliminated? This would be a necessary action to restore the ability of the immunological synapse to function again.

Response: In our AFM experiments, the dysfunctional, PS-bound perforin shows heights above the membrane that are inconsistent with any known transient state of perforin pore formation, and is found to be resistant to washing with buffer (including EGTA, thus removing calcium from the solution). We therefore deduce that it cannot be regenerated and will remain permanently inactive, as per the reviewer comment above.

Our experiments on CTLs do not provide an answer to the fate of PS-bound perforin after synapse formation. As a consequence, we do not have experimental evidence to demonstrate whether PS-bound perforin may affect the ability of the immunological synapse to function again, nor about its subsequent elimination (by endocytosis or other means). However, elimination via endocytosis is certainly a possibility, as we have shown previously that membrane-bound perforin becomes internalised through constitutive membrane endocytosis (Ref 15). Furthermore, the fact that CTLs are capable of serial killing (as per Ref 8) strongly suggests that the released perforin is unlikely to interfere with synapse formation.

4. The authors state, in reference to Fig. 5, starting on line 178 that “Taken together, these observations on PS are consistent with perforin assembly that is trapped in a dysfunctional (dead-end) state upon its initial insertion into the membrane, failing to complete the prepore-to-pore transition.” I think that the phrase “initial insertion into the membrane” is misleading, as it implies a dysfunctional insertion of the β -barrel pore. It appears that perforin “binds” to the PS membrane but cannot convert to a closed ring. Therefore, binding to the PS appears to prevent the monomers from undergoing the conformational changes necessary to the proper interaction of monomers, which changes the geometry of the interaction so that the monomers are stuck a more open, curvilinear configuration and cannot convert to a circular pore.

Response: As shown in our earlier work on perforin (Leung et al., Nat. Nano 2017) and as has been demonstrated for various other pore forming proteins (e.g., as reviewed in Hodel et al., Curr Opin Struct Biol 2016), curvilinear assemblies can form transmembrane pores and closed-ring formation is not a prerequisite for pore formation. In addition, perforin does show some degree of oligomerisation on PS, with a subunit spacing and assembly size that is similar to that of perforin prepores (see Fig. 5). Yet the reviewer is correct in noting that the PS may prevent the subsequent tighter monomer interaction/packing that was previously found to coincide with the prepore-to-pore transition (Leung et al., Nat. Nanotech 2017). We have therefore changed the wording of this sentence to “...perforin assembly that is trapped in a dysfunctional (dead-end) state that prevents it from completing the prepore-to-pore transition.”

5. In Fig. 4 the authors need to denote the colors in the figure legend for panel 4a, right-most panel. Also, the increased height of the unlocked prepores of 5 nm is fairly substantial, it is 40% of the height of the perforin molecule. Do the authors have any

explanation for this quite dramatic increase in height? It is hard to envision how this happens based on what we know about the mechanism.

Response: The colour scale has now been indicated at the end of the caption of Fig. 4 (it is the same for the AFM images in Fig. 2 and in Fig. 4b,c, for consistency). The height increase on PS is indeed substantial. We have validated that height increase extensively and found similar maximum heights with perforin on (negatively charged) DOPG membranes (Fig. S7 in the manuscript). We could speculate that it may be due to some monomers attempting to extend the transmembrane hairpins but failing to insert into the membrane. As a consequence, the protein may be pushed up with respect to the membrane surface. However, since we do not have structural data for the dysfunctional state observed on PS (and PG) lipids, we prefer to refrain from such speculation in the manuscript. As pointed out by the reviewer, such a structural change is outside of what we know of perforin pore formation, which leads us to argue that it is a dysfunctional, perhaps denatured, “dead-end” state of perforin.

Reviewer #2 (Remarks to the Author):

Cytotoxic cells form an immune synapse with their target cells. Into the confined space of the immune synapse, the cytotoxic cells then release perforin that forms pores in the target cell's plasma membrane. Now, apoptotic serine proteases (granzymes) enter the target cell and initiate apoptosis. The cytotoxic cell is not affected, despite being exposed to its own secretion into the cleft of the immunological cleft. The manuscript addresses, how the cytotoxic cells protect themselves against their own toxic secretions.

The authors argue that the protection is afforded by prevention of pore formation by perforin. They claim two mechanisms. Firstly, they show that perforin requires a disordered lipid phase to form pores in vitro, in AFM. They show that a reduction in membrane order of cytotoxic cells by loading with a cholesterol analogue that does not support lipid raft formation highly increases lysis of these cells by perforin. They show a moderate increase in this respect for target cells that are already readily lysed. They further show a role of phosphatidylserine. They show in vitro in both TEM and AFM that PS disrupts pore formation and leads to clusters. They show that perforin in CTLs co-localizes with PS. They claim that in the immunological synapse, PS is introduced to the outer leaflet of the CTL membrane and prevents pore formation.

Assessment:

In my opinion, this is a well executed study that demonstrates the possibility that lipid organization is responsible for the protection of cytotoxic cells from their own toxic secretion. Cell biological- and biophysical methods are well designed and executed. Data analysis is adequate. Overall, I am excited to see attention paid on the role of lipids for this important biological function and I am excited about this paper overall.

Strength (this is what I believe the authors have proven):

- that the perforin binds much less to CTLs than target cells and that these cells are therefore protected.*
- that disruption of lipid rafts reverses this protection.*
- that perforin forms pores only in the lipid disordered phase of phase separated membranes.*
- that PS or other negatively charged membrane lipids sequester perforin into dysfunctional aggregates rather than pores.*

Response: We thank the reviewer for this positive assessment of the manuscript, and here provide a point-by-point response to his specific comments below.

In my assessment, the conclusions that CTL are protected by a high-order lipid membrane and that PS is externalized by CTLs as an additional protective mechanism are not yet fully supported by the evidence. The authors have proven that this is a possibility. Specifically, the authors do not show that the overall lipid order in CTLs is greater than in target cells. This should be easy for them, using the lipid order sensor laurdan. This is significant for this manuscript, in my opinion. A great number of membrane functions is associated with lipid rafts. Most membrane proteins are either raft- or non-raft proteins and may loose- or change their function if the membrane order is altered. Disrupting rafts changes or abolishes protein-dependent and -independent functions alike. Seeing that pore formation is likely a simple self-assembly process and is apparently not dependent on other proteins I still believe that the authors have a strong point, particularly when also considering their work on model membranes. It is however important to strengthen this point by comparing CTLs to target cells. As well, the GP changes quite moderately: does the change support the dramatic effect the authors have observed in terms of pore formation?

Moreover, does the moderately lower GP after treatment in CTLs equate the status in target cells pre-treatment? Unless these additional experiments provide a clear-cut prove, I would put the lipid-order based interpretation of the results forth as a strong possibility rather than a certainty.

Response: We agree with the reviewer that a direct comparison of membrane order of EL4 target cells and CTLs would be ideal. However, to our best knowledge, there are no previous studies directly comparing membrane order (using laurdan) of different cell types, in particular for such vastly different cells as the cancer cell lines and primary cells tested here. There are various reasons for why such a comparison might prove to be difficult (if possible at all). In our view the most significant issue precluding such a comparison is that different cell types have different amounts and localisation of intracellular (very low order) membranes. Many of these are in close proximity to the plasma membrane, to within the optical resolution of our experiments. Therefore, despite best measures being taken to avoid inclusion of these internal membranes (Ref 51), the measured GP at the plasma membrane will contain a contribution from intracellular membranes too. This contribution will be different between different cell types, given differences in how close to each other the plasma and intracellular membranes are in different cell types. As a consequence, such a laurdan comparison will not be a reliable probe for differences in plasma membrane order between different cell types.

That said, although we are unable to provide a direct laurdan GP comparison, we believe the reviewer has still made an important point. We, therefore, now provide additional data (Figure R2, included now as Figure S6 in the manuscript) showing a vastly (over 30-fold) higher level of GM1 (as indicated by CTxB staining) on the CTL plasma membrane compared to that found on EL4 cells. GM1 has been used extensively in the literature as a marker of lipid rafts, which in turn have been associated with high membrane order (Refs 31,32). We believe these experiments support the presence of a clear-cut difference in lipid order between CTLs and target cells, fully consistent with our observations and conclusions.

Figure R2: GM1 surface staining by recombinant cholera toxin subunit B Alexa 647. GM1 intensity (represented by CTxB) has been plotted against CD8+ positivity to identify CD8+ T cells in a 1:1 mixture of EL4 and OTI T cells (cells are gated for same size as detailed for perforin binding experiments in manuscript). Average MFI (\pm SD) of CTxB-Alexa 647 from 3 independent experiments is included for both CD8 positive and negative cells.

With respect to “moderately lower GP” (Figure 3A), since in our experiments we aimed to analyse the effects of a cytolytic protein following 7KC loading, we had to ensure that the 7KC induced changes in membrane order were not cytotoxic, at least during the course of our experiments, i.e., 2 hours (Figure 3b). To our knowledge, 7KC treatment of CD8+ T lymphocytes has not been performed before (the majority of studies have focused on CD4+ T cell lines). It could be that the GP values observed here represent the

maximum tolerated level of membrane disorder in this cell type. As noted above, we cannot use laurdan GP to directly compare plasma membrane order between CTLs and target cells (instead, we used CTxB to label lipid rafts; Figure R2 and new Figure S6).

Finally, all of these findings are consistent with the disruption of membrane order (by 7KC) in model membranes (Figure S5).

Regarding the role of PS: Does PS co-localize with perforin in target cells as well (i.e. is this a consequence of membrane association of perforin)? This should be established as a control. The video S3 shows PS in the contact zone of the two cells. Whether PS is on one or the other cell or on both is not resolved. Therefore, the evidence is not strong enough to proclaim this a "mechanism", in my opinion and may be better described as a possibility. Further experiments may clarify this point.

Response: To determine whether perforin and PS also co-localise on the EL4 target cells, we used a similar experimental set up and analysis as that shown in Fig. 6a,b for T cells; the only difference between the two experiments was that here we omitted CTxB due to the very low signal as shown by flow cytometry in Fig. R2 (new Figure S6), and we had to use a lower concentration of perforin to minimise cell lysis. It was found that, similar to T cells, there was a high co-localisation between PS and WT-GFP-PRF signal on EL4 cells (Figure R3a,b).

It is important to emphasise that these experiments (both the initial CTL co-localisation analysis in the manuscript, and also the new EL4 data) were specifically designed to avoid imaging artefacts which could arise from PS externalisation ("flip-flop") caused by sub-lytic concentration of perforin as has been shown in (Ref. 30 in manuscript). Thus, in our experiments here and in the original manuscript (as described in *Preparation and imaging of fixed cell samples for colocalization analysis* in Materials and Methods), the cells were exposed first to annexin V Alexa 568, washed, then treated with WT-GFP-PRF, washed again, fixed and analysed by microscopy. Therefore, only PS present on the cell surface prior to WT-GFP-PRF addition would be labelled with annexin Alexa 568. Any PS that could be externalised following perforin interaction with the membrane (and leading to abnormally high co-localisation values) would not be labelled.

To further confirm that this experimental design avoids detection of any PS "flip/flop", flow cytometry analysis was performed on cells prepared in the same way: i.e., stained with annexin V Alexa 568, washed, and then exposed to WT-GFP-PRF (Figure R3c). Whereas the GFP signal representing perforin binding clearly increases in a dose dependent manner, the annexin signal remains stable across the entire sub-lytic range (including the concentration used for the microscopy analysis). Hence, the PS-PRF co-localisation shown in Figures 6a,b is not a consequence of membrane modification by perforin. To clarify the importance of this experimental design, we have now added the following sentence to the manuscript: 'Importantly, this pre-labelling with Annexin V ensured detection only of PS already exposed on the CTL membrane before perforin addition, thus avoiding any contribution of PS 'flip/flop' upon perforin association with the plasma membrane ³⁰.'

Figure R3: Perforin binding to phosphatidylserine on the ELA plasma membrane

- Colocalization of externalised PS (detected by annexin V Alexa 568 and recombinant WT-GFP-PRF (signal enhanced by GFP TAG polyclonal antibody, Alexa 488) on an ELA cell. The nucleus (shown in blue) is detected by Hoechst 34580.
- Quantitative analysis (as described in the methods) of a, showing high co-localisation of PS with WT-GFP-PRF (78%). A total of 50 cells were analysed, and the mean of 3 independent experiments is shown (\pm s.e.m.)
- Flow cytometry analysis of both CTL (triangles) and ELA (circles) which have been pre-labelled with annexin V Alexa568 before incubation with titrating amounts of WT-GFP-PRF. A clear increase in GFP signal is observed as perforin concentration increases, however the MFI of the annexin V Alexa 568 does not increase, demonstrating that preparing cells in this way does not detect PS exposure due to membrane “flip-flop”.

We agree with the Reviewer that our results (Video S3 and Fig 6d) do not conclusively determine the origin of the PS signal within the synapse. However, there is a plethora of published data demonstrating PS enrichment on the presynaptic membrane within the immune synapse, including mass spectrometry (Ref 37). A mechanism responsible for PS enrichment on the pre-synaptic T cell membrane has also been defined: continuous Ca^{2+} signalling by the T cell inhibits flippase activity and leads to PS externalisation on the T cell membrane (Fischer et al., Ref 31). Moreover, and importantly, our experiments shown in Figure 6d and Video S3 were performed using PRF-knockout T cells (we have now clarified this in the manuscript). Therefore, upon synapse formation (as confirmed by centrosome docking), no perforin was released into the synaptic cleft and no perforin-induced “flip-flop” of the target cell membrane was possible. Of course, we cannot rule out some other unknown factors that may influence target cell membrane asymmetry, but we feel that our manuscript – combined with the previous literature – provides overwhelming direct and indirect evidence for PS exposure on the pre-synaptic membrane.

Overall, this is an exciting manuscript.

Matthias Amrein

Reviewer #3 (Remarks to the Author):

This is a very interesting and highly novel paper that convincingly demonstrates that the biophysical properties of plasma membrane protects cytotoxic T lymphocytes (CTLs) from perforin. The experiments are well performed and I have only minor issues:

- In the abstract, the authors state that the resulting resistance of CTLs to perforin explains the ability to kill target cells in rapid succession. Is the CTL membrane remodeled before and after perforin release or is phosphatidylserine permanently at elevated levels in the outer leaflet of the plasma membrane?*

Response: A number of studies suggest that the CTL membrane is remodelled (e.g. increased CTL membrane order within the synapse as per Refs 34, 35) before degranulation/perforin release. PS enrichment within the immune synapse appears to be constitutive, as continuous Ca^{2+} signalling suppresses flippase activity and leads to PS externalisation (Fischer et al. Ref 31). Furthermore, Zech et al. (Ref 37) have demonstrated PS and SM/cholesterol (lipid rafts) enrichment within the immunological synapse using mass spectrometry. Interestingly, our data using unconjugated T cells (Figure 6a,b,c and Video S1 and S2) also demonstrate a level of constitutive exposure of PS on the CTL membrane in the absence of synapse formation (of course, one caveat of this proposition is that these cells potentially formed a synapse during their activation from the naïve state, but this is virtually impossible to confirm). Currently there is no evidence indicating whether the PS exposed within the synapse is “permanent” or not. To answer this question, long-term single cell analysis will be required; while this is an interesting problem, it falls outside the scope of this manuscript.

- The membrane order is higher in the periphery of the immunological synapse compared to the center of the synapse (Ref 33 and 34 [Ref 34 and 35 in this resubmission]). Would the authors expect more perforin binding to the center of the synapse and have they observed this in CTL synapses?*

Response: Gaus et al (Ref 34) demonstrated using multiple experimental systems that membrane order at the effector cell synapse rapidly increased within 3-7 min in the centre of the synapse. At a significantly later stage, by ~20 min and in some instances by 60 min, it begins to decrease moving towards a ‘ring’ shape around the peripheral synapse. Indeed, the authors state that ‘...within the IS, condensed membranes are first formed in central regions and are then rearranged to more peripheral sites’.

Other studies have demonstrated that the formation of the immune synapse leads to Ca^{2+} influx within 5 minutes (eg. Le Borgne et al., J Immunol 2016, or Lioudyno et al., PNAS 2008), and perforin release follows shortly thereafter. Thus, in our own work (Refs 8 and 15), we have shown (see relevant figures below, from Ref 8) that both human and mouse primary T and NK cells release perforin (as indicated by “PI blush” in the target cell - below) within 2-5 min of Ca^{2+} flux in the CTL, and effector lymphocytes remain invariably protected from perforin (no “PI blush” - below). Thus, the invariable protection of the effector lymphocytes from perforin (as shown by us previously and further investigated in the current manuscript at the molecular level) strongly suggests that at the time of perforin release, the membrane order would still be highest in the central synapse area. This will preclude perforin binding to the CTL membrane, other than where PS is exposed. Owen et al (Ref 35) demonstrated high membrane order at the periphery of the synapse only, but in that study the authors started taking images 10 min after synapse formation, and one can deduce that similar observations to those reported in Ref 34 would have been made had they started imaging the synapse at an earlier time-point.

Figure R4 (Ref. 8, Figure 1): Timing between calcium flux in mouse CD8+ T cell, and 'PI blush' observed in SIINFEKL labelled target cells. 'PI Blush' is observed in the target cell (146 seconds) but not in the effector cell. Quantification of this data shows calcium signalling within minutes of stable synapse formation, and PI blush in the target cell within minutes of calcium signalling.

- *In the experiment shown in Figure 4, are the charged lipids required per se for perforin assembly or are the charged lipid required to create a surface potential? In other words, can assembly be blocked by increasing the ionic strength in solution?*

Response: We have considered other experiments that modify the membrane surface potential, e.g., by varying the concentration of monovalent or divalent ions (i.e., change ionic strength as the reviewer suggests), or by varying pH. The main problem is that this could affect perforin binding, assembly and pore formation in multiple ways (Voskoboinik et al., Nat Rev Immunol 2015). Firstly, perforin pore formation is known to be strongly dependent on pH, with the low pH in the secretory granules retaining perforin in an inactive state during its intracellular transport to the presynaptic membrane. Secondly, perforin binding critically depends on the presence of calcium ions, and changing the concentration of other ions is likely to affect the interaction between calcium and perforin too. Finally, by thus changing surface potential of the membrane via the ionic strength in solution, one would also change the surface potential of perforin itself, likely affecting the oligomerisation interface. In other words, such experiments are unlikely to provide a conclusive answer to the reviewer's question. However, given that the same effect is observed on membranes that differ in chemical composition but that are similar in surface charge/potential (Fig. S7), we attribute the effect of PS to the more generic membrane surface potential instead of to the specific lipid.

- *An interesting but optional experiment would be to induce phosphatidylserine to the outer leaflet of the plasma membrane in target cells and examine whether elevated phosphatidylserine levels are sufficient to prevent CTL killing.*

Response: The only feasible ways of inducing PS exposure (above the basal level) in target cells are initiating apoptosis or interference with a range of proteins responsible for the plasma membrane asymmetry. However, this will initiate other (difficult to account for) cellular and plasma membrane perturbations, which will make it difficult to interpret the results. However, this is certainly an interesting suggestion that is worth exploring further in future studies.

Our AFM data showing non-porating perforin oligomers on PS membrane (Figures 4, 5 and S7) addresses this question in an artificial setting. As discussed in the manuscript, an irreversible (non-porating) binding of WT-PRF-GFP to CTLs further supports this notion (Figure 1).

I would like to emphasize that all of the proposed experiments are optional (but may improve the manuscript); they are not required for publication.

Reviewers' Comments:

Reviewer #1:

Remarks to the Author:

The authors have adequately replied to the previous comments by this reviewer and I have no further comments

Reviewer #2:

Remarks to the Author:

The issues noted by this reviewer as well as other reviewers have been adequately addressed. I recommend publication.

Matthias Amrein

Reviewer #3:

Remarks to the Author:

The authors have answered my questions.